# The comparison of cancer gene mutation frequencies in Chinese and U.S. patient populations

Fayang Ma[1,2], Kyle Laster [2] & Zigang Dong [1,2] ✉

Knowing the mutation frequency of cancer genes in China is crucial for reducing the global health burden. We integrate the tumor epidemiological statistics with cancer gene mutation rates identified in 11,948 cancer patients to determine their weighted proportions within a Chinese cancer patient cohort. *TP53* (51.4%), *LRP1B* (13.4%), *PIK3CA* (11.6%), *KRAS* (11.1%), *EGFR* (10.6%), and *APC* (10.5%) are identified as the top mutated cancer genes in China. Additionally, 18 common cancer types from both China and U.S. cohorts are analyzed and classified into three patterns principally based upon *TP53* mutation rates: *TP53*-Top, *TP53*-Plus, and Non-*TP53*. Next, corresponding similarities and prominent differences are identified upon comparing the mutational profiles from both cohorts. Finally, the potential population-specific and environmental risk factors underlying the disparities in cancer gene mutation rates between the U.S. and China are analyzed. Here, we show and compare the mutation rates of cancer genes in Chinese and U.S. population cohorts, for a better understanding of the associated etiological and epidemiological factors, which are important for cancer prevention and therapy.

Cancer is one of the leading causes of death for humans. In 2020, an estimated 19.3 million new cancer cases and 10.0 million cancer deaths occurred worldwide[1]. China ranked highest with 4.6 million new cases and 3.0 million cancer deaths, accounting for 24% of newly diagnosed cases and 30% of cancer deaths globally[2]. For the U.S., it is estimated that 1.9 million new cases and 0.6 million cancer deaths have occurred in 2021. However, the cancer death rate has decreased continuously between 1991–2018, which is largely attributed to smoking cessation initiatives and advances in detection and treatment modalities[3].

There are commonalities and differences in cancer incidence rates worldwide and identification of population-specific etiologies may serve to decrease the global health burden. Measures for effective cancer prevention can be enacted only when the underlying etiology is determined. In China, HBV vaccines have dramatically reduced the prevalence of HBV infection, a major risk factor for liver cancer[4]. As a result, the incidence and mortality rates of liver cancer have decreased since the late 1970s[5]. Additionally, China exhibits the highest regional incidence rates of esophageal squamous cell carcinoma (ESCC) for both men and women. Potential risk factors for ESCC include consumption of high-temperature drinks and food, and pickled vegetables. However, the incidence rates of ESCC are broadly in decline due to improved living standards in China. In Western countries, the reduction of ESCC incidence is considered primarily due to a large-scale decline in cigarette smoking[6]. For certain types of cancer, the exact underlying etiology has not been fully elucidated. For example, African-Americans and Asians living in Korea and Japan (but not in the U.S.) had higher death rates from lung cancer than individuals of European descent[7]. Lung cancer incidence rates were higher and more variable among women in East Asia (EAS) than in other geographic areas with low female smoking rates[7]. Factors other than cigarette smoking may mainly account for the increasing incidence of lung adenocarcinoma (LUAD) among EAS, European (EUR) and U.S. females[8–12].

[1]Department of Pathophysiology, School of Basic Medical Sciences, College of Medicine, Zhengzhou University, Zhengzhou, China. [2]China-US (Henan) Hormel Cancer Institute, Zhengzhou, China. ✉e-mail: dongzg@zzu.edu.cn

Genomic sequencing has provided an increasingly comprehensive view of the genetic mutations implicated in tumorigenesis and has paved the way for the development of personalized cancer therapies[13,14]. Additionally, multi-national comparisons are critical to determine differences in racially or regionally biased mutated cancer genes. In a recent U.S. pan-cancer study, *TP53* (34.5%), *PIK3CA* (13.5%), *LRP1B* (13.1%), *KRAS* (10.5%), *APC* (10.1%), *FAT4* (9.5%), *KMT2D* (9.2%), *KMT2C* (9.1%), *BRAF* (7.7%), and *ARID1A* (7.0%) were identified as the top mutated cancer genes for the U.S. population[15].

In this work, by integrating genomic and epidemiological data, we calculated the weighted proportions of observed mutated cancer genes within the Chinese population. We then compared the weighted cancer gene mutation rates between the Chinese and U.S. pan-cancer datasets. The differences observed in cancer gene composition and mutation rates are potentially attributable to genetic predisposition, ethnic diversity, population-specific and environmental risk factors. These investigations are significant for guiding cancer prevention and drug development.

## Results

### Prevalence rates of mutated cancer genes in China
Epidemiological data documenting the incidence rates of cancer within the Chinese population were derived from the 2004-2016 editions of the "China Cancer Registry Annual Report"; details in Methods. According to cancer types in the epidemiological information, publicly available sequencing profiles of Chinese cancer patients were collected from the cBioPortal data repository and supplemented with several other sources (see Methods). The disparity between the cancers observed within the epidemiological data and the sequencing data should be noted; several cancers, including laryngeal, testicular, and vaginal cancers, are under-represented within the sequenced datasets. Thus, we overlapped the cancer types represented in the two datasets and exclusively focused on the anatomical sites at the intersection. In total, 94 detailed cancer subtypes (11,948 cancer patients/samples) with both qualified epidemiological information and sequencing profiles were included for analysis (Fig. 1). The corresponding demographic and clinical data of the patients included within our investigation are shown in Supplementary Fig. 1a–f. The regions of sample origin were overlapped with cancer registry location presented within the "China Cancer Registry Annual Report" (Supplementary Fig. 2a). The mutation data was then filtered for quality assurance. In the tier-1 filtering, the mutation profiles of 94 cancer subtypes were filtered using a census panel of 382 cancer genes to facilitate identification of mutated cancer genes within each cancer subtype. In tier-2 filtering, the results from the tier-1 filtering were further processed to retain only protein coding mutations that confer amino acid substitutions. The Mutation Profiles_1 of the 94 cancer subtypes were organized and reclassified into Mutation Profile_2 of 23 major tumor sites based on the ICD_10 classification system (Supplementary Data 1_Sequencing data_23 mutation profiles). To quantify the mutation proportion of cancer genes across the 23 major cancers represented in the China pan-cancer dataset, we integrated the mutation rates with epidemiological cancer statistics to produce a weighted percentage of the CN_382 mutated cancer genes. We identified that *TP53* (51.4%), *LRP1B* (13.4%), *PIK3CA* (11.6%), *KRAS* (11.1%), *EGFR* (10.6%), and *APC* (10.5%) are the top mutated cancer genes in Chinese cancer patients. The ranked prevalence of mutated cancer genes can serve as guidance for etiology investigation, cancer prevention, and drug development.

### Comparison of cancer gene mutation rates between CN and US
The comparison of cancer mutation profiles across population-based groups is critical to determine if there are differences in racially or regionally biased mutated genes. Mendiratta et al.[15] recently produced a list containing 21,271 gene mutation proportions observed within the U.S. pan-cancer based on epidemiology data (ICD-O-3) spanning nearly two decades. However, the epidemiological data of Chinese cancer patients were collected based on the ICD-10 classification system via nationally distributed cancer registry centers. To ensure an equal comparison of mutation rates between CN cohorts and U.S. cohorts, we recalculated the U.S. mutations rates by reclassifying the categories presented in the ICD-O-3 format to the ICD-10 format (see Methods).

The comparison of the epidemiologically weighted mutation rates of the 382 cancer genes derived from CN pan-cancer datasets and U.S. pan-cancer datasets showed no statistically significant differences ($p = 0.2014$) (Fig. 2a). The mutation rates of the 382 cancer genes were highly and positively correlated ($r = 0.95$) (Fig. 2b); the degree of correlation between the two datasets was slightly decreased after the contribution of *TP53* was removed ($r = 0.93$) (Fig. 2c). Interestingly, we noticed that *EGFR* deviated the most from the best-fit line: *EGFR*_CN (10.6%) vs *EGFR*_U.S. (3.1%). The differences in *EGFR* mutation rates between the two populations are further investigated in Fig. 5.

The top 50 genes from the CN_382 and U.S._382 cancer gene lists (Supplementary Data 1_Sequencing data_MutationFrequencies_382)

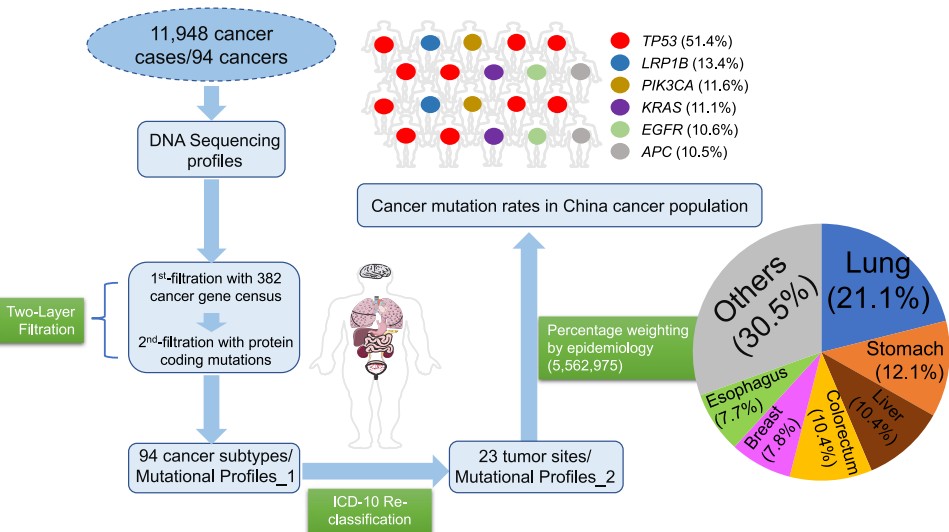

**Fig. 1 | Workflow of current study.** The workflow for calculating epidemiologically weighted cancer gene mutation proportions within the Chinese cancer population. Source data are provided as a Source Data File.

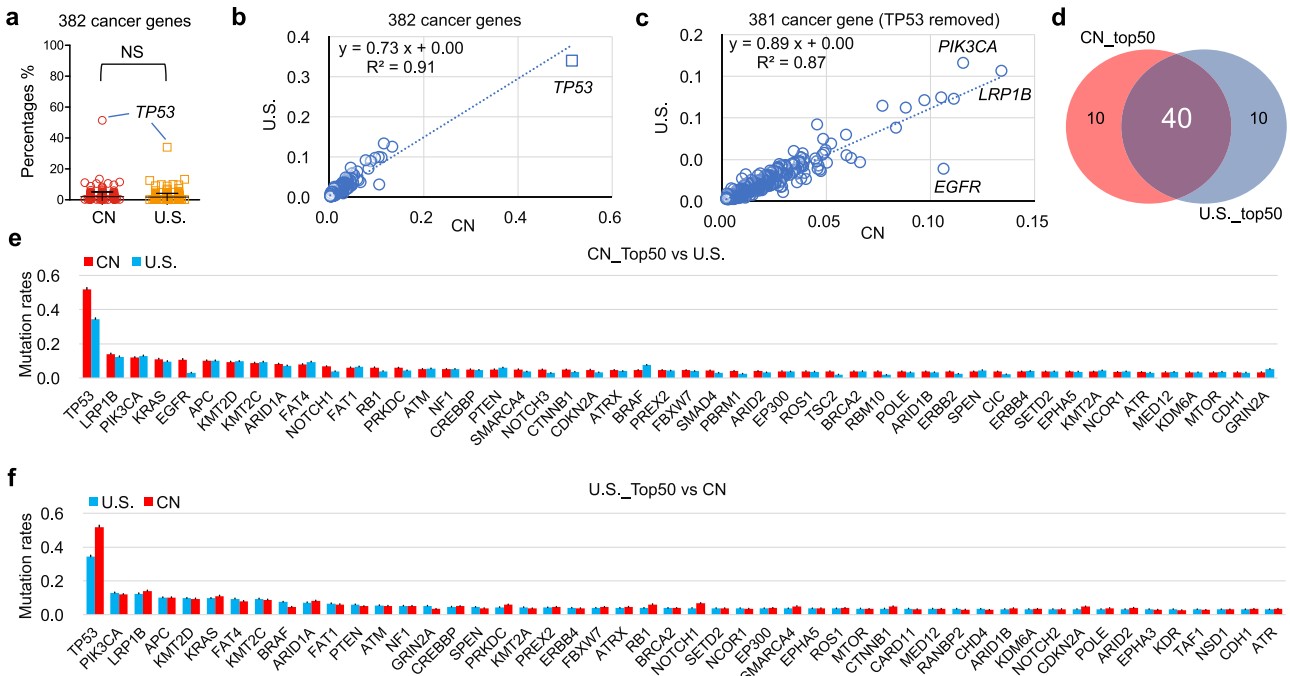

**Fig. 2 | The comparison of the mutation rates of the top50 cancer genes between CN and U.S. patient population. a** The comparison of the epidemiologically weighted mutation rates of the 382 cancer genes derived from CN pan-cancer datasets ($n = 11,948$ patients) and U.S. pan-cancer datasets ($n = 18,584$ patients), $p = 0.2014$ (unpaired $t$ test with Welch's correction, two-tailed, Welch-Corrected $t = 1.279$, $df = 713$; $F$ test: $F = 1.707$, $DFn = 381$, $Dfd = 381$, $p < 0.0001$), CN _ Mean ± SE = 0.02067 ± 0.001578 ($n = 382$ genes), U.S. _ Mean±SE = 0.01813 ± 0.001208 ($n = 382$ genes), 95% confidence interval −0.001354 to 0.006437. **b** The correlation analysis visualizing the epidemiologically weighted mutation rates of the 382 genes observed in CN and U.S. patient populations, the blue dashed line indicates the linear trend, Pearson $r = 0.9542$, 95% confidence interval 0.9443 -

0.9624, $p < 0.0001$. **c** The correlation analysis detailing the mutation rates of the 381 genes (the dominant contribution of *TP53* was removed) between the two cohorts, Pearson $r = 0.9307$, 95% confidence interval 0.9159 to 0.9430, $p < 0.0001$. **d** 40 of the top50 mutated cancer genes were shared between CN and U.S. pancancer datasets. **e–f** Comparison of the weighted mutation frequencies of the top50 cancer genes derived from CN ($n = 11,948$ patients) and U.S. ($n = 18,584$ patients) pan-cancer datasets, respectively. Mean±Error, error bars represent the 95% confidence limits determined through simulated samples ($n = 2000$ independent Poisson distributed computational samples with the calculated mutation proportion as the central value), and measure of centre (bar levels) represent mean of simulated mutation proportions. Source data are provided as a Source Data File.

were compared to evaluate the similarity rate of the significantly mutated cancer genes. We found that 80% (40 / top 50) were shared between Chinese and U.S. patient populations (Fig. 2d–f). The top 10 mutated cancer genes observed in the weighted China pan-cancer dataset were *TP53* (51.4%), *LRP1B* (13.4%), *PIK3CA* (11.6%), *KRAS* (11.1%), *EGFR* (10.6%), *APC* (10.5%), *KMT2D* (9.7%), *KMT2C* (8.8%), *ARID1A* (8.4%), and *FAT4* (7.7%). The top mutated cancer gene is *TP53* both in China (51.4%) and U.S. (34.0%). The mutation rates of the remaining cancer genes were relatively low in both populations, with the majority at frequencies below 10%, and the overall mutation rates of cancer genes in China and U.S. are generally equivalent.

## Top mutated cancer genes derived from CN and U.S. cohorts

18 cancer types with high prevalence in either China or the U.S. were selected. Next, the most prevalent mutated cancer gene in each cancer type was determined from the corresponding mutation profiles for both population groups. Finally, the most prevalent mutated gene in each of the 18 cancer types for both population groups were summarized (Fig. 3a, b). The results indicated that *TP53* is ranked highest in nine cancer types of the CN cohort (51.4%) and in eight cancer types of the U.S. cohort (34.0%). This observation is unsurprising but indicates the reason the weighted mutation proportion of *TP53* within the cancer populations is so high. *BRAF* is the top mutated gene in the skin cutaneous melanoma (SKCM) and papillary thyroid carcinoma (THCA/PTC) cohorts in both countries. In most cancer types included in this study, U.S. and CN patients share the same top gene; *KRAS* for pancreatic adenocarcinoma (PAAD) in CN & U.S., *PIK3CA* for cervical squamous cell carcinoma (CESC) in CN & U.S., *EGFR* for lung adenocarcinoma (LUAD) in CN, *APC* for colorectal adenocarcinoma (COCA)

in U.S., *PTEN* for uterine corpus endometrial carcinoma (UCEC) in CN & U.S., *IDH1* for brain glioblastoma and low grade glioma (GBM) in CN & U.S., *VHL* for kidney renal clear cell carcinoma (KIRC) in CN & U.S., *SPOP* for prostate adenocarcinoma (PRCA) in CN & U.S. *TP53*, *KRAS*, *PIK3CA*, *EGFR*, *APC*, *BRAF*, *PTEN*, *IDH1*, *VHL*, and *SPOP*, as the top mutated cancer genes in different cancer types, are further analyzed in the following results.

## Three mutational patterns were identified in CN and U.S. cohorts

The mutation rates of the top 50 genes between the China and U.S. patient cohorts were compared among the 18 most common cancer types (Fig. 4a–c). The results showed that *TP53*, *KRAS*, *PIK3CA*, *APC*, *PTEN*, and *BRAF* are frequently mutated in most of the cancer types analyzed. *TP53*, a tumor suppressor gene, is the most frequently mutated gene in nearly all cancers[16,17]. Based upon the *TP53* mutation rate and rank in each respective cancer cohort, we classified the 18 cancer types into three distinct patterns: *TP53*-Top, *TP53*-Plus, and Non-*TP53* (Fig. 4a–c). In the *TP53*-Top pattern, *TP53* is ranked highest and is the most prevalent mutated gene in ESCC, OVCA, LUSC, HNSC, GACA, BLCA, and LIHC in both China and U.S patient cohorts (Fig. 4a). The highest mutation rate of *TP53* was observed in the OVCA_CN cohort (93.5%) and the lowest mutation rate in the LIHC_U.S. cohort (26.4%). Our findings indicated that *TP53* has the highest mutation rates in the China and U.S. pan-cancer datasets and is the top mutated cancer gene in nearly half of the 18 most common cancers.

COCA, LUAD, BRCA, PAAD and GBM were classified into the *TP53*-Plus pattern (Fig. 4b), in which *TP53* is clustered with one/two other

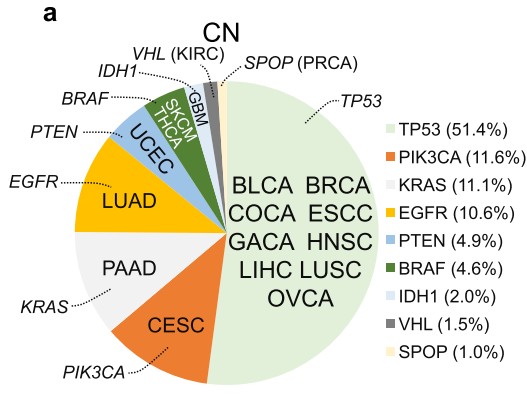

**Fig. 3 | Top mutated genes in CN and U.S. cancers. a, b** The top mutated genes for each of the 18 cancer types in the China and U.S pan-cancer cohorts. The case number for each cohort are BLCA (CN_163 vs U.S._411), BRCA (CN_303 vs U.S._1020), CESC (CN_76 vs U.S._289), COCA (CN_1541 vs U.S._545), ESCC (CN_914 vs U.S._95), GACA (CN_973 vs U.S._439), GBM (CN_286 vs U.S._896), HNSC (CN_94 vs U.S._508), KIRC (CN_243 vs U.S._361), LIHC (CN_1131 vs U.S._364), LUAD (CN_1370 vs U.S._1027), LUSC (CN_392 vs U.S._485), OVCA (CN_185 vs U.S._426), PAAD (CN_461 vs U.S._177), PRCA (CN_65 vs U.S._497), PTC (CN_71 vs U.S._346), SKCM (CN_27 vs U.S._366), and UCEC (CN_49 vs U.S._531). Cancer type abbreviations: Esophageal squamous cell carcinoma (ESCC), High-grade serous ovarian carcinoma (OVCA), Lung squamous cell carcinoma (LUSC), Head and neck squamous cell carcinoma (HNSC), Gastric adenocarcinoma (GACA), Bladder urothelial cancer (BLCA), Liver hepatocellular carcinoma (LIHC), Colorectal adenocarcinoma (COCA), Lung adenocarcinoma (LUAD), Breast cancer (BRCA), Pancreatic adenocarcinoma (PAAD), Brain glioblastoma/glioma (GBM), Papillary thyroid carcinoma (PTC), Skin cutaneous melanoma (SKCM), Kidney renal clear cell carcinoma (KIRC), Uterine corpus endometrial carcinoma (UCEC), Cervical squamous cell carcinoma (CESC), Prostate adenocarcinoma (PRCA). Source data are provided as a Source Data File.

genes with respect to mutation rate. In the COCA, BRCA, PAAD and GBM patient cohorts from China and the U.S., the observed gene clusters are *TP53/APC/KRAS*, *TP53/PIK3CA*, *KRAS/TP53* and *IDH1/TP53*, respectively. However, distinct top mutated gene clusters were observed in the LUAD_CN (*EGFR/TP53*) and LUAD_U.S. cohorts (*TP53/EGFR/KRAS*). Recently, a genomic analysis of 103 Chinese LUADs showed that *EGFR* (50%) and *TP53* (51%) are predominantly mutated genes and that co-mutation of *EGFR* and *TP53* often indicated poorer prognoses than those harboring *EGFR* mutations alone[18].

The *KRAS/TP53* gene cluster is the most highly ranked in the Chinese and U.S. PAAD cohorts. Previous studies have shown that *KRAS*, *TP53*, *CDKN2A*, and *SMAD4* are the four major driver genes identified in pancreatic cancer. Of these four driver mutations, genetic alternations in *KRAS* and *CDKN2A* have been suggested as early events in pancreatic tumorigenesis[19]. The *IDH1/TP53* gene cluster is the most highly ranked in the GBM-CN/U.S. cohorts. GBM is the most common primary malignant brain tumor. A comprehensive analysis of 22 GBM samples led to the discovery of recurrent mutations in the active site of isocitrate dehydrogenase 1 (*IDH1*) in 12% of GBM patients. These genetic alterations are potentially useful for the classification and targeted therapy of GBMs[20]. Recently, it has been proposed that glioblastoma be reclassified based on molecular profiling, with particular emphasis placed on IDH mutation status[21].

In the Non-*TP53* pattern, the mutation rates of several genes exceed those observed with respect to *TP53*. The Chinese and U.S. PTC, SKCM, KIRC, UCEC, CESC, and PRCA cohorts were grouped into this pattern (Fig. 4c). The top mutated gene for PTC_CN/U.S and SKCM_CN/U.S. cohorts is *BRAF*. *VHL*, *PTEN*, *PIK3CA*, and *SPOP* are the top mutated genes of Chinese and U.S. KIRC, UCEC, CESC, and PRCA cohorts, respectively.

In the UCEC cohorts, *PTEN* (69.4% in UCEC_CN, 53.7% in UCEC_U.S.) and *PIK3CA* (34.7% in CN, 49.2% in U.S.) are the top two mutated genes. The tumor suppressor gene *PTEN*, a negative regulator of the *PI3K/AKT/mTOR* pathway, is mutated and lost in up to 80% of UCEC tumors. *PTEN* mutations frequently coexist with mutations in *PIK3CA*, *PIK3R1*, and *KRAS* within a given UCEC tumor[22]. The mutation signature is concordant with type I endometrioid carcinomas, which are preferentially associated with mutations in *PTEN* (52%–78%), *KRAS*, *CTNNB1* and *PIK3CA*, whereas type II serous carcinomas frequently harbor *TP53* (60%–91%) mutation and *HER2* amplification[22]. *PIK3CA* is

the top mutated gene for the CESC_CN (35.5%) and CESC_U.S. (29.4%) cohorts. Additionally, *KMT2C/D* and *FBXW7* mutations were frequently observed in both populations. Most CESCs are characterized with APOBEC signature mutations, predominantly caused by human papillomavirus (HPV) infection, and suggests that APOBEC activity is a key driver for *PIK3CA* mutagenesis and HPV-induced transformation[23]. *BRAF* is the top mutated gene for the PTC and SKCM cohorts of both populations. Papillary thyroid cancer (PTC) is the most common type of thyroid cancer. Previously, *BRAF^V600E* and *RAS* mutations were identified as the most common pathogenic mutations with respect to PTC. Interestingly, the observation of general mutual exclusion between *BRAF*, *NRAS*, *HRAS*, and *KRAS* mutations confirmed the crucial role of *MAPK* signaling alterations in PTC carcinogenesis[24]. Activation of *BRAF* in response to mutation has been suggested as the earliest and most common genetic alteration in human melanoma[25]. *VHL/PBRM1* is the top mutated gene cluster in the KIRC cohorts of both patient populations. Kidney cancer is driven by metabolic alterations, and *VHL* mutations dysregulate tumor response to such changes[26]. In a survey comprised of 400 clear cell renal cell carcinoma samples, *VHL* (controlling cellular oxygen sensing) and *PBRM1* (maintaining the chromatin states) were identified as significantly mutated[27]. *SPOP* is the top mutated gene in the PRCA_CN and PRCA_U.S. cohorts. Tumors harboring *SPOP* mutations have the highest levels of androgen receptor-induced transcripts. Based on the mutation status of *SPOP* and five other genetic mutations, 74% (246/333) of primary PRCAs were classified into seven subtypes[28].

The top 50 genes in each of 18 cancer types from the Chinese and U.S. cohorts were overlapped to determine the common mutated gene rates between the two populations. We found large variances within the top 50 mutated genes across the 18 cancer types of both populations, with common mutated gene rates in the top 50 genes ranging between 28% and 100% (Fig. 4d). GACA (100%), COCA (84%), LUSC (78%), LUAD (70%), UCEC (70%), and BLCA (64%) are the 6 cancers with the highest commonly mutated gene rates. The PTC (28%) and SKCM (36%) cohorts have the lowest rates. It can be assumed that the mechanisms contributing to carcinogenesis in both populations would be similar if the same driver gene mutations are shared between them. Additionally, we found that the average mutation rate observed in the PTC_CN cohort is significantly higher than the average rate observed in the PTC_U.S. cohort. On the contrary, the average mutation rate

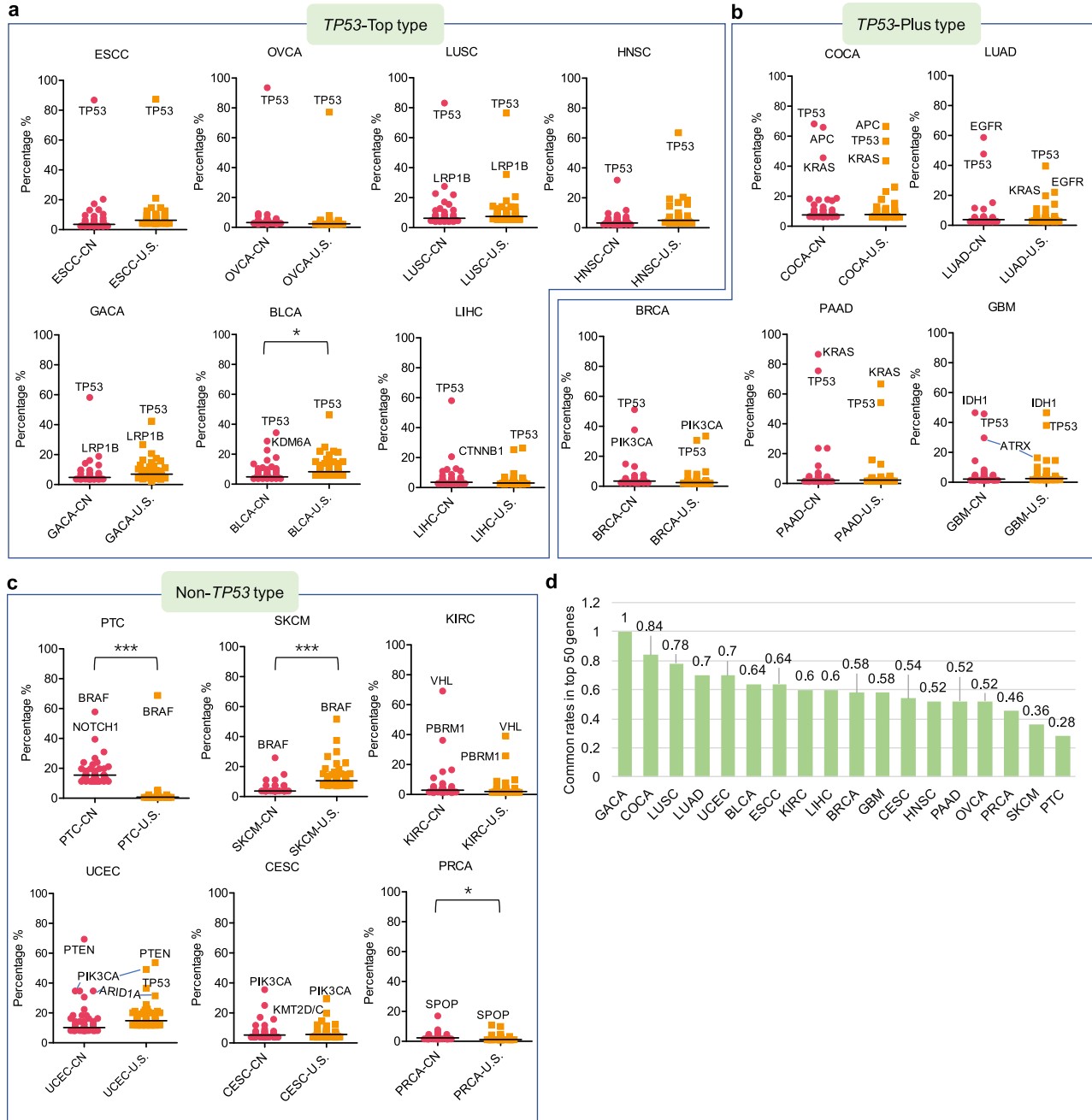

**Fig. 4 | The classification of 18 common cancer types into three patterns.**
**a**–**c** The comparison of the mutation rates of top 50 cancer genes in each of the 18 most common solid tumor types from China and U.S. cohorts. The 18 cancers are classified into three types: *TP53*-Top type, *TP53*-Plus type, and Non-*TP53* type, based on the rank of *TP53* with respect to other top mutated genes. The differences between China and U.S. cohorts were statistically evaluated using the unpaired *t*

test. * represents 0.01 < *p* < 0.05, ** represents 0.001 < *p* < 0.01, *** represents *p* < 0.001, two-sided, 95% confidence interval. The cancer type abbreviations could be referred to Fig. 3 legend. **d** The top 50 genes within each cancer type from both populations were overlapped to determine the corresponding common rate in the top 50 genes, which is calculated in 2* (TotalGeneNumber – TotalUniqueGeneNumber) / TotalGeneNumber. Source data are provided as a Source Data File.

observed in the SKCM_U.S. cohort is significantly higher than the average mutation rate observed in the SKCM_CN cohort. These differences were the most statistically significant across the 18 cancer types analyzed. The potential etiological factors underlying these differences were analyzed below.

### *EGFR* and *TP53* mutation rates comparison between CN and U.S. cohorts

*TP53* is the top mutated cancer gene identified in Chinese and U.S. cancer patient populations. The mutation rates of *TP53* in the 20 cancer cohorts from China and U.S. were summarized and compared

(Fig. 5a). The mutation rates of *TP53* ranged between 0.9% (PTC_U.S.) ~ 93.5% (OVCA_CN) (Fig. 5b). Our analysis indicates that the increased *TP53* mutation rates in the Chinese cancer patient populations was largely attributed to the higher mutation rates of *TP53* in LUSC_CN, GACA_CN, ESCC_CN, LIHC_CN, and LUAD_CN after being integrated with the corresponding cancer epidemiology data (Supplementary Data 3_Epidemiological data_CN_ICD10).

The mutation rate of *EGFR* in Chinese cancer patients is 10.6%, which is higher than that observed in the weighted U.S. pan-cancer (3.1%) estimates. We found that *EGFR* (58.7%) is predominantly mutated at higher frequencies in CN_ LUAD compared to 22.1% in

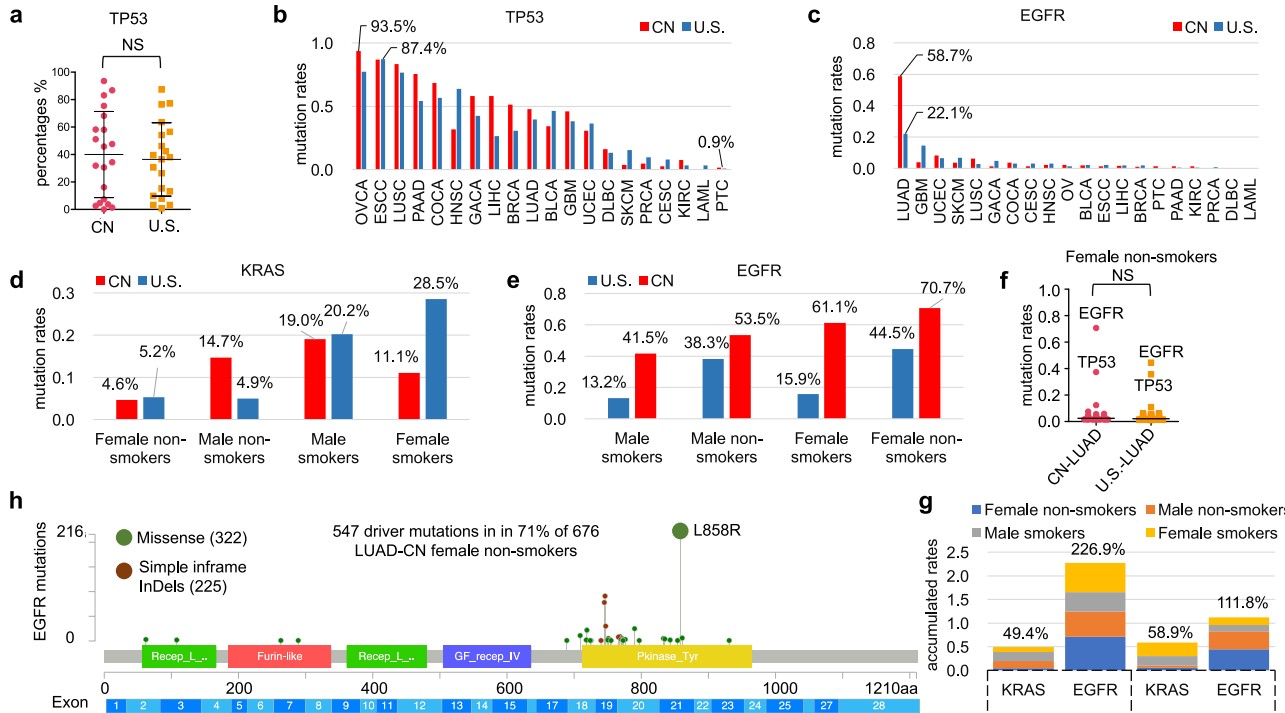

**Fig. 5 | The comparison of *TP53* and *EGFR* mutation rates between CN and U.S. a, b** The mutation rates of *TP53*, derived from the 20 most common cancer types, were compared between CN and U.S., respectively. $p = 0.6994$ (unpaired $t$ test, two-tailed, $t = 0.3891$, $df = 38$; $F$ test: $F = 1.379$, $DFn = 19$, $Dfd = 19$, $p = 0.4899$), CN _ Mean ±SE = 0.4003 ± 0.07017 ($n = 20$), U.S. _ Mean ± SE = 0.3644 ± 0.05975 ($n = 20$), 95% confidence interval −0.1508 to 0.2225. **c** The mutation rates of *EGFR* were visualized and compared in 20 cancers from CN and U.S. cohorts. **d, e** In LUAD, the mutations rates of *KRAS* and *EGFR* between Chinese and U.S. patients were compared with respect to gender and smoking status. **f** Comparison of the top 50 mutated genes

from Chinese ($n = 676$) and U.S. ($n = 191$) female LUAD non-smokers, the difference was not statistically significant (NS), $p = 0.7221$ (unpaired $t$ test with Welch's correction, two-tailed, $t = 0.3568$, $df = 88$; $F$ test: $F = 2.007$, $DFn = 49$, $Dfd = 49$, $p = 0.0163$), CN _ Mean±SE = 0.05003 ± 0.01532 ($n = 50$), U.S. _ Mean ± SE = 0.04334 ± 0.01081 ($n = 50$), 95% confidence interval −0.03064 to 0.04402. **g** The accumulated mutation rates of *KRAS* (**d**) and *EGFR* (**e**) in CN and U.S. cohorts, respectively. **h** The spatial distribution of *EGFR* mutations in Chinese lung adenocarcinoma patients as female non-smokers visualized with a lollipop graph. Source data are provided as a Source Data File.

U.S._LUAD and other cancer types in both populations (Fig. 5c). To determine the etiological or demographical factors responsible for the disparity, the mutational profiles of LUAD patients within each population were stratified according to gender and smoking-status (Supplementary Data 1_Sequencing data_LUAD Sex-Smoking Status). However, we consistently observed higher *KRAS* mutation rates (except male non-smokers) in U.S. patients than in Chinese patients (Fig. 5d), and higher *EGFR* mutation rates in Chinese patients than in U.S. patients irrespective of gender or smoking status (Fig. 5e). Of all the demographics analyzed, the Chinese LUAD_female non-smokers (70.7%) exhibited the highest *EGFR* mutation rates (Fig. 5e). However, no statistically significant differences were observed in the top 50 cancer gene mutation profiles between China and U.S. LUAD_female non-smokers (Fig. 5f). Additionally, we observed much higher *EGFR* accumulated mutation rate in CN-LUAD patients (226.9%) than that in U.S. (111.8%). For *KRAS*, the accumulated mutation rate between CN (49.4%) and U.S. (58.9%) is almost comparable (Fig. 5g).

In total, 547 *EGFR* driver mutations were identified in the 478 Chinese LUAD_female non-smokers. The *EGFR*[L858R] mutation occurred in 45.2% (216/478) of these patients (Fig. 5h). It is estimated that 40-55% of Asian NSCLC patients have tyrosine kinase inhibitor (TKI)-sensitive mutations, with *EGFR*[L858R] and *EGFR*[Ex19Del] comprising most cases[29]. In contrast, 16.4% (45/275) of U.S. female TCGA_LUAD patients have *EGFR* mutations, and only 6.2% (17/275) of these patients harbor the *EGFR*[L858R] mutation. These observations suggest the existence of other unknown environmental/cultural risk factors that may contribute to the elevated *EGFR* mutation frequency observed in Chinese LUAD cancer patients.

## Etiological factors for CN and U.S._LUAD, SKCM, and PTC

Cancer somatic mutations are largely the product of repair processes which are tightly associated with DNA replication. Distinctive patterns of mutational signatures are indicative of these replication and repair processes sustained over the course of tumor development[30]. We utilized the SignatureAnalyzer program to generate mutation spectra detailing single nucleotide variants occurring within the top 50 mutated cancer genes of the LUAD, SKCM, and PTC patient cohorts before executing mutation signature prediction through non-negative matrix factorization. Only samples with treatment-naïve status were retained for the mutational signature analysis (Supplementary Data 2_Treatment status data). The signature contribution for the analyzed cancers are presented below and the corresponding cosine similarity plots were provided in Supplementary Fig. 5. The results showed that the mutation signature for female non-smokers in CN_LUAD and U.S._LUAD were SBS40, an unknown etiology mutation signature (Fig. 6a, b). The potential underlying etiological factors are discussed in the discussion section.

Our previous analysis of the mutation rates observed between the SKCM and PTC patient cohorts of both populations indicated that their differences were among the most statistically significant (Fig. 4c). We identified differences within the mutation signatures for the SKCM cohorts of the Chinese (SBS7b) and U.S. (SBS7a) patient populations (Fig. 6c, d). However, the proposed etiology of both signatures fell in the category of ultraviolet radiation exposure, which has been associated with DNA replication timing[30]. Additionally, the SBS7a/b signatures were mutually exclusive with the normal aging related mutational signature SBS1/5[31,32]. The mutational signatures for PTC_CN and PTC_U.S. cohorts were both SBS25 (Fig. 6e, f), which is possibly

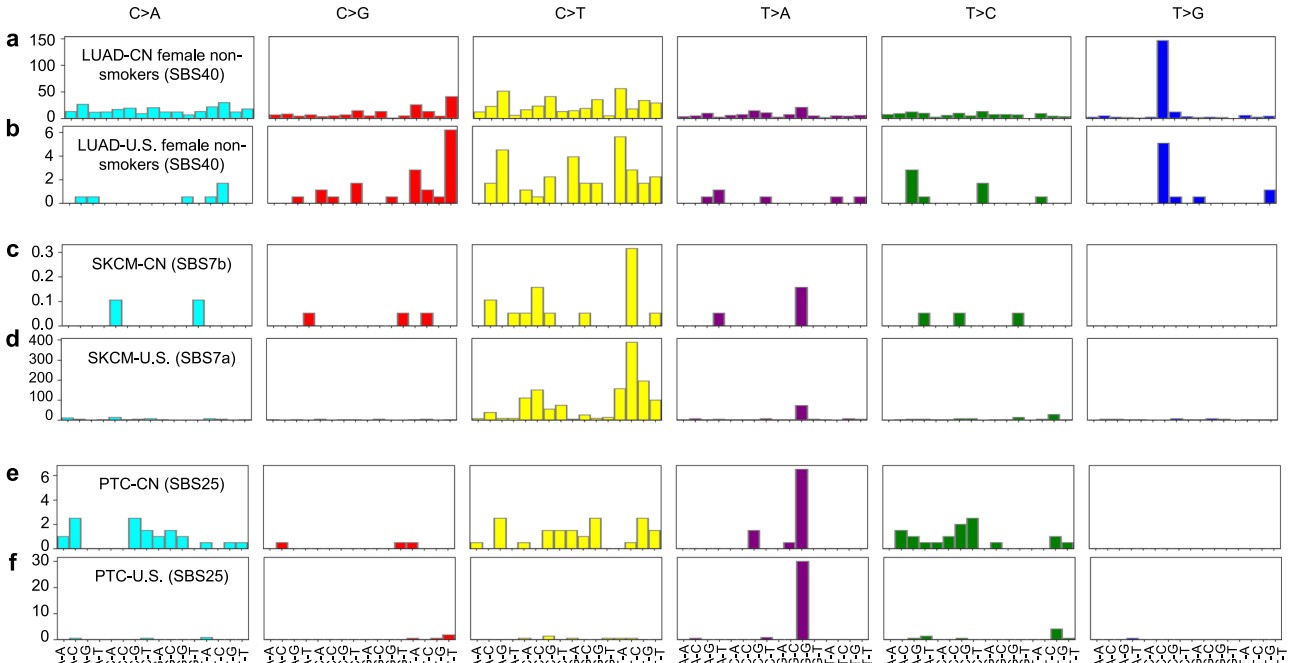

**Fig. 6 | Mutational signature analysis of lung adenocarcinoma, skin cutaneous melanoma, and papillary thyroid carcinoma in CN and U.S. a, b** The mutation signature derived from the mutational spectra across the top 50 mutated genes in China (LUAD_CN, $n = 550$) and U.S. (LUAD_U.S., $n = 57$) female non-smokers. **c–f** The mutation signatures of skin cutaneous melanoma (SKCM_CN, $n = 21$; SKCM_U.S., $n = 225$) and papillary thyroid cancer (PTC_CN, $n = 67$; PTC_U.S., $n = 104$) derived from the mutational spectra of the top 50 mutated genes in the Chinese and U.S. patient cohorts, respectively. Source data for mutational signature analysis are provided in Supplementary Software 1.

linked to chemotherapy treatment. The inconsistency between the treatment-naïve status (included samples) and possible chemotherapy treatment (produced mutational signatures) indicated other potential etiological factors, which require further investigation and identification. However, it was reported that the low somatic mutation density in thyroid cancer relative to other cancers is correlated with age and not associated with genotype or radiation exposure[24], which partially explains the significantly lower mutation rates observed in the U.S. PTC cohort relative to the Chinese PTC cohort.

## Discussion

Carcinogenesis is a dynamic, complex process caused by the interplay between genetic susceptibility and environmental factors, resulting in variable phenotypes across ethnicities and geography. It is estimated that there are nearly 140 driver genes that contribute to cancer development[33]. The generally accepted consensus is that cancer is initiated and progresses due to alterations in between two to eight critical driver genes[33]. Driver mutations with overwhelming carcinogenicity were observed in *TP53*, the most frequently mutated tumor suppressor gene in human[34]. This implies that the classification of cancer based upon driver mutation clusters, such as those containing *TP53* mutations, is a reliable means to identify tumors that may benefit most from targeted gene therapies. Additionally, for a certain cancer, the shared top mutated cancer genes between populations likely suggest common etiological factors. However, the variability within population-level mutation profiles with respect to gene constitution and mutation frequency may suggest varied mechanisms responsible for driving carcinogenesis.

In the present study, 382 cancer-associated genes (from OncoKB database) mutated in 18 solid cancers were investigated and compared between China and U.S. cancer populations. We found that most of the analyzed cancer types between Chinese and U.S. cohorts share the same top mutated driver gene or gene cluster. However, various driver mutations within a single gene[35–38] or across different genes[39] have

differing carcinogenic potential and can produce synergistic effects with respect to tumor progression. A deeper understanding of tumorigenesis resulting from driver mutations will likely be achieved as the cost of single-cell sequencing is reduced and bioinformatic approaches continue to mature.

In China and other East Asian (EAS) countries, *EGFR* mutations were found in 30%–50% of LUAD patients[10,18,40]. In contrast, *EGFR* mutation rates were only observed in between 8%–21% of LUAD patients from the U.S. and other regions[18,40–42]. *EGFR* mutations were identified at a rate of 59.4% in female Asian LUAD patients with no history of smoking[43]. On the other hand, *KRAS* mutations were found in 6%–11% of Chinese LUADs[10,18] compared with 25%–33% in U.S and other populations[18,41,42]. However, *TP53* mutation rates in Chinese (36%–51%) and U.S. (46%) LUAD cohorts were nearly equivalent[10,18]. *TP53* and *LRP1B* mutations were concurrent in EAS cohorts, whereas *KRAS* and *EGFR* mutations were mutually exclusive[10]. Additionally, integrative genomic analysis has shown that mutation of *EGFR* and *TP53* generally occur prior to whole genome doubling and most local somatic copy number alterations[44,45]. Co-mutation of *EGFR* and *TP53* was associated with poor survival in LUAD patients, which was partially attributed to the aggressive nature of their tumors[18,46], thereby highlighting a crucial role of these two driver mutations in the *EGFR*-mutant EAS LUAD population. In summary, LUADs harboring *EGFR* and *TP53* mutations are enriched in EAS female non-smokers, while most LUADs with *TP53* and *KRAS* mutations tend to occur in male smokers of European descent[10,40].

A genomic study reported by the TCGA revealed a unique mutational landscape in tumors derived from Chinese NSCLC patients compared to those from patients of European descent[47], suggesting genetic diversity of the cancer genome between ancestries[48]. In detail, EAS LUADs are characterized by more stable genomes with fewer mutations and copy number alterations at the chromosomal level, lower driver abundance and tumor mutation burden, and a higher proportion of intratumor heterogeneity due to early genomic

diversification in evolutionary trajectory. In contrast, tumors from the EUR cohorts were largely characterized by multiple genomic alterations attributed to chronic tobacco exposure[10,45,49,50].

Epidemiologically, among cigarette smokers, African-Americans and Native Hawaiians are more susceptible to lung cancer than Hispanic, Japanese-American, and those of European descent[51]. In China, increased rates of lung cancer in men reflect high smoking rates but increased rates among non-smoking women appear to be related to other factors[52]. In our analysis, the disparity of *EGFR* mutation rates between Chinese and U.S. LUAD patients remained after stratifying the cohorts according to gender or smoking status. Genetically, the SNPs significantly associated with increased lung cancer risk in the Chinese population are different from those identified in patients of European descent[53]. It has been suggested that ancestry is one of the key factors responsible for the genetic differences between the two populations. In several Asian populations, including China, an increased risk of lung cancer in women is also likely associated with indoor pollution produced by heating and cooking oil fumes[52,54]. Importantly, household pollutants resulting from coal and biomass fuel combustion have been classified as Group 1 and Group 2 A carcinogen for lung cancer by the IARC, respectively[55]. Taken together, intrinsic genetic predisposition[53] or environmental factors, especially air pollution and cooking oil fumes[56,57], can play important roles in the regional differences observed in lung cancer incidence rates.

Thyroid cancer incidence is increasing worldwide, with the largest number of cases occurring in China and the U.S.[58,59]; however, the underlying etiology has not yet been elucidated. Potential risk factors for thyroid cancer include radiation exposure, increased iodine intake, and environmental pollutants such as nitrates and heavy metals[60]. Recently, an investigation conducted in the U.S. found that nearly two-thirds of large papillary thyroid cancer was correlated with increased body weight and obesity[61]. Additionally, increased body weight and obesity are also partially responsible for the high incidence of endometrial carcinoma worldwide, especially in countries that have undergone rapid socio-economic transitions, such as China[62]. In the U.S., the estimated incidence of skin melanoma ranks fifth across all cancers in both genders, with a total of 106,110 cases in 2021[3]. In contrast, only 7714 skin melanoma cases occurred in China (Globocan 2020). Despite the large difference in the incidence rate of skin melanoma between countries, over-exposure to ultraviolet radiation in the form of sunlight is the principal risk factor. Therefore, melanoma could be effectively minimised in both populations by limiting sun exposure[63].

ESCC incidence rates show remarkable variation worldwide and are not fully explained by known lifestyle and environmental risk factors[6]. The risk factors for ESCC include lifestyle factors (smoking, alcohol drinking, and consuming pickled food or high temperature drink/food), poor diet, and genetic susceptibility[64]. Recently, a multi-national comparative study of 552 ESCC genomes indicated that the mutation profiles were similar across all the investigated countries[65]. Our results also showed no significant differences in the mutation rates of cancer genes of ESCC between the Chinese and U.S. population groups. Additionally, Chinese and U.S. ESCC patient cohorts were grouped in the *TP53*-Top pattern with common mutation rate as high as 64% (Fig. 4d). Evidence accumulated from areas with high ESCC incidence rates, including China, Iran, South America, and East Africa, suggests that consumption of hot food/beverages (exceeding 65ºC) is likely the leading etiological risk factor for ESCC[66]. Therefore, ESCC incidence rates may be potentially curbed by allowing food and drink to cool before consumption[67].

Ancestry differences observed in cancer incidence rates have long been suggested as contributors to many tumor types. Key examples of differences include genetic predisposition[68,69], environmental factors, and distinct lifestyles[70–72]. The evidence presented above suggests that the environmental factors accounting for lung adenocarcinoma, esophageal squamous cell carcinoma, skin melanoma, papillary thyroid carcinoma, and endometrial carcinoma are likely the same across China and the U.S., although mutagen exposure likely varies in duration and intensity. As we enter the era of precision cancer medicine, understanding the varied mutation rates of cancer genes across all tumor types in different countries will undoubtedly lead to more efficient targeted therapies. A concerted multi-national effort is crucial for overcoming barriers and balancing geographical disparities in research and health care delivery[73].

## Methods
### Epidemiological Data
In order to quantify the mutation proportions across tumors derived from Chinese patients, we first determined cancer incidence rates within the Chinese population. Epidemiological data for calculating cancer incidence rates between 2004–2016 were collected (Supplementary Data 3_Epidemiological data_CN_ICD10). The annual cancer incidences of 2004–2016 were cited from the "China Cancer Registry Annual Report" series, which were tabulated and published by China National Cancer Center (NCC). According to the "Guidelines of Chinese Cancer Registration", "Technical Protocols of Cancer Registration and Follow Up", and the standards of International Agency for Research on Cancer/International Association of Cancer Registries (IARC/IACR) on "Cancer Incidence in Five Continents, Vol. XI", a national criterion to evaluate the quality of Chinese cancer registration data was established following the rules of comparability, completeness, validity, and timeliness.

Cancer registries collected data on all cancers' incidence, mortality, and survival. The demographic information and diagnostic information of the registered cancer cases were also recorded. Cancer incidence was recorded following the standards of ICD-O-3 or ICD-10. Next, these records were reclassified based on the ICD-10 classification system. The detailed methods used for data collection in cancer registries, including Methods and Index, Methods of Data Collection, Channels of Data Collection, Certification of Cancer Cases, and Follow-up Practice, were provided in China Cancer Registry Annual Report, 2004–2016. After receiving the cancer registration data, NCC utilized the IARC/IACR-check software to evaluate the completeness, validity, and internal consistency of the data. The cancer datasets were examined and revised based on the evaluation results. Finally, qualified cancer datasets were pooled and published for annual national cancer report. For instance, a total of 1,110,867 cancer patients registered in 487 qualified cancer registries were included in the "China Cancer Registry Annual Report, 2019", which covered a total of 381,565,422 population (193,632,323 males, 187,933,099 females), accounting for 27.6% of the national population in 2016 (Supplementary Data 3_Epidemiological data_CN_registry data 2004-2016/487 registries). These cancer registries are representative of nearly all provinces in China (Supplementary Fig. 2a). In the 13-year range between 2004–2016, 5,878,712 cancer patients were registered (Supplementary Fig. 2b, c). The population coverage data, as a function of census data, was collected from departments of statistics and public security. In which, Han Chinese comprised 91.59%, 91.51%, and 91.11% of the total Chinese population in 2000, 2010, and 2020, respectively (Data from National Bureau of Statistics, http://www.stats.gov.cn/tjsj/tjgb/rkpcgb/). The proportions of urban population in the total population were documented as 36.22%, 49.68%, and 63.89% in 2000, 2010, and 2020, respectively (Supplementary Fig. 2d) (Data from National Bureau of Statistics). Over the past decades, the increased speed of urbanization and population flow at the national level resulted in greater genomic hybridization and homogenization in China (Supplementary Fig. 2e).

For the U.S. cohorts, the SEER 18 registries (2000–2017) are commonly used in epidemiological statistical analysis (https://seer.cancer.gov/statistics-network). Available registry data includes the Alaska Native Tumor Registry, Connecticut, Detroit, Georgia Center

for Cancer Statistics (Atlanta, Greater Georgia, Rural Georgia), Greater Bay Area Cancer Registry (San Francisco-Oakland, San Jose-Monterey, Greater California, Hawaii, Iowa, Kentucky, Los Angeles, Louisiana, New Mexico, New Jersey, Seattle-Puget Sound, Utah) (Supplementary Data 3_Epidemiological data_U.S._epidemiology and samples). The cancer registry data covered 27.7% of the U.S. population according to the race and ethnicity statistics represented within the SEER registry. Delay-adjusted incidence rates for European Descent and African Descent appear comparable over a 19-year period. An increase in the number of American-Indian/Alaskan Native is apparent, while a decrease can be seen for the Hispanic population. A flat-trend is observed for the Asian/Pacific Islanders. Importantly, the statistical analyses presented in SEER utilize a technique known as race bridging which allows data collected using one set of race categories consistent with data collected using a different set of race categories. The bridged-race estimates used within the study were produced under a collaborative arrangement with the U.S. Census Bureau. Corresponding details could be found at https://www.cdc.gov/nchs/nvss/bridged_race.htm. The ancestry influences on genomic profiles have also been considered and provided for the included U.S. cohorts. From the 139 studies included in the U.S. cohorts (Supplementary Data 3_Epidemiological data_U.S. Ancestry_sequenced), 11,526 cancer patients with known ancestry background were presented in the pie chart (Supplementary Fig. 2f).

## Cancer Genomic Data Processing

Publicly available cancer genomic sequencing data of the Chinese population were collected from cBioPortal (OrigiMed2020)[74,75], ICGC, Chinese Glioma Genome Atlas[76], brain cancer[77], lymphoma[78,79]. The genomic data of U.S. cancer patients were derived from cBioPortal ("MSK-IMPACT Clinical Sequencing Cohort", and "Cancer Therapy and Clonal Hematopoiesis"), the ICGC data portal, and TCGA (MAF files provided by the BROAD Firebrowse database). For the CN cohorts, 11,948 cases in 94 detailed cancer subtypes were included for analysis. The corresponding 94 mutation profiles underwent a two-tier filtration scheme. The tier-1 filtering was achieved by retaining all cancer genes within the OrigiMed2020 dataset that are represented in the Memorial Sloan Kettering Precision Oncology Knowledge Base (OncoKB; 382 genes)[80]. The tier-2 filtering was achieved by retaining protein-coding mutations, including missense mutations, nonsense mutations (truncating mutations), short inframe InDels, and splice mutations, which could contribute to changes in the protein sequence. Structural variants (SV) and gene fusions are not included in the results. Next, the filtered 94 mutation profiles corresponding to 94 detailed cancer subtypes were integrated into 23 mutation profiles matched to 23 major cancer types according to the ICD-10 classification scheme used in the cancer epidemiological data collection. As gene names may vary depending on the selected sources, we utilized an in-house Mathematica script to map any gene synonyms to the official gene name.

## Performance evaluation of different variant callers

Large differences in variant calls have been reported when the same sequencing data was processed with variant multiple calling algorithms[81]. However, our main focus was to calculate the proportion of cancer patients harboring a specific mutated gene (from a panel of 382 cancer genes) in one cohort, the cancer patient would be counted if one or more harbored mutations fall into our established category. With this consideration in mind, the requirement of strict consistency among the results produced from different variant callers markedly decreases. Publicly available sequencing data was used to test the consistency of the variant calls produced from commonly used variant callers in the current study. The four commonly used variant callers (Mutect2, Varscan2, Somaticsniper, and Muse) were used for variant aggregation and masking against the GDC TCGA Esophageal Cancer. The corresponding mutation profiles were downloaded

(Supplementary Data 4_Variant caller data_VariantCallerDataLink) and analyzed in the procedure (Supplementary Fig. 3a). The consistencies were analyzed in terms of Patient, Gene, Gene-Mutation, and Gene-Patient via overlapping the corresponding lists produced from the four variant callers (Supplementary Fig. 3b-e). We found that 81.1% (231/285) of the four mutated gene lists produced from the variant callers were commonly shared. If two variant callers were used in a comparison, that percentage would be greatly increased to 82.3%–94.3%. Additionally, the patient number corresponding to each of the shared mutated genes produced from the four variant callers were compared and showed no significant differences (Supplementary Fig. 3f). Most importantly, the patient number corresponding to each of the commonly shared mutated genes produced from the four callers were well overlapping with each other (Supplementary Fig. 3g). With respect to Supplementary Fig. 3g, we developed a metric to assess the maximum error exist within this particular dataset. 7.7% as an maximum error rate was resulted from calculating the sum of the absolute value of pairwise differences in observed variant call for each individual gene across the four variant callers. Above evidence demonstrated that the mutation rates of cancer genes produced from these variant callers were generally consistent and not significantly different from each other. The corresponding lists produced by the four variant callers are presented in the Supplementary Data 4_Variant callers data.

## Conversion of mutations rates from ICD-O-3 into ICD-10

The U.S. mutation rates of cancer genes were epidemiologically weighted based on the ICD-O-3 classification system. In our analysis, these mutation rates were recalculated after reclassification of ICD-O-3 into ICD-10 classification system, to ensure the equal comparison of CN and U.S. mutation rates of cancer genes on the same standard. As such, the cancer incidence across all histological subtypes within that tissue were summed and divided by the total cancer incidence to determine the preliminary weight of each tissue site. Based upon the preliminary weights, the percentage of tumors without corresponding sequencing data were calculated as value Q. The final epidemiological weights were calculated through rescaling, specifically by dividing the preliminary weights of the tumors with sequencing data by (1 - value Q). 66 cancer subtypes (18,584 cases) were clustered and re-allocated across 23 tumor sites after reclassification of ICD-O-3 into ICD-10 (Supplementary Data 1_Sequencing data_Cases and subtypes). To test the accuracy and consistency of the epidemiologically weighted mutation rates of genes produced based ICD-O-3 and ICD-10, the panel of these 382 cancer genes used for filtering in CN cohorts was utilized to be compared between the two U.S. datasets derived from ICD-O-3 and ICD-10. No statistically significant differences ($p = 0.7964$) were observed between the ICD-10 and ICD-O-3 produced mutation rates with respect to the 382 cancer genes (Supplementary Fig. 4a). Additionally, the mutation rates of the 382 cancer genes were highly and positively correlated (Supplementary Fig. 4b), even after the removal of TP53 (Supplementary Fig. 4c). The top50 of the 382 cancer genes produced from both classification systems were visualized and showed rather equivalent mutation rates (Supplementary Fig. 4d, e).

## Mutation proportion estimates

Mutation proportion estimates were calculated as described in the publication[15]. Briefly, the **m** x **n** matrix was constructed, detailing each cancer gene (**m**) and the number of cancer types sequenced (**n**). The entries within the matrix represent the conditional probability that gene $m_i$ is mutated in cancer type $n_j$; importantly, a gene was tabulated as mutated a maximum of one time per sample despite the existence of 1 or more observed mutations. Within the present study, the number of genes (m) is 382 and the number of cancer types (n) is 23. The epidemiological cancer incidence rates were converted into a numerical vector **v** representing their percentage contribution to the total number of cancer cases. The final mutation proportion of each gene

was calculated by taking the scalar product of the values in each row **m**, which details the conditional probability that a gene is mutated across the n cancer types, and the values in vector **v**. Statistical analyses were performed by generating two thousand Poisson distributed in silico datasets for each combination of gene and cancer type with the mean value as calculated from the genomic studies included. The pipeline used for the initial mutation proportion estimate was applied to the in silico datasets, and the 95% confidence intervals were determined based on these calculations.

## Tumor mutation signatures

Mutation signatures were produced by using the SignatureAnalyzer program developed by the Getz laboratory[82]. The mutated nucleotide sequence of each gene derived from genomic data were utilized to determine the respective COSMIC SBS signature for a specific cancer type. The Hugo Gene Symbol, Patient ID #, Chromosome, Mutation Start Position, Reference Allele, Variant Allele, and Variant Type were parsed from the patient mutation data and concatenated into maf files for each individual cancer type within the present study. Spectra files for each cancer type were then generated with the signature-analyzer.spectra.get_spectra_from_maf command using cosmic3 and the hg19.2 bit reference genome assembly as the input arguments. The signatureanalyzer.run_spectra command was then used to generate the mutation signatures using cosmic3 and nruns=10 as the input arguments. To verify that the mutational spectra derived from top 50 mutated genes of a cancer type meets the threshold for signature identification, we queried the corresponding data derived from lung adenocarcinoma from U.S. patients (LUAD_US) with smoking history. The use of this input produced a mutational signature (SBS4) for smoking, indicating that the spectra derived from the top 50 genes is reasonable and reliable to discern relevant mutation signatures.

## Statistics

Unpaired $t$ test was used to compare the differences between two independent datasets. $F$ test to compare variances, if $p < 0.05$, unpaired $t$ test with Welch's correction would be used to compare differences of means. The detailed statistics information of each statistical comparison (unpaired $t$ test with/without Welch's correction, $p$ value, $t$, $df$, Mean±SE, 95% confidence interval; $F$ value, $DFn$, $Dfd$, $p$-value) could be found in the corresponding figure legend. GraphPad Prism 5 and Microsoft Excel were used for all the statistical calculation and figure production. An asterisk symbol * represents $0.01 < p < 0.05$, ** represents $0.001 < p < 0.01$, *** represents $p < 0.001$, two-tailed, 95% confidence interval. All measurements were taken from distinct samples and not measured repeatedly.

## Reporting summary

Further information on research design is available in the Nature Research Reporting Summary linked to this article.

## Data availability

All the data used in this work are publicly available and no new data were generated in this study. The genetic mutational data are publicly available [https://www.cbioportal.org/], [https://dcc.icgc.org/], [http://www.cgga.org.cn], [https://pubmed.ncbi.nlm.nih.gov/25171927/], [https://pubmed.ncbi.nlm.nih.gov/32183952/], [http://gdac.broadinstitute.org]. The cancer epidemiological data of 2004-2016 are derived from the "China Cancer Registry Annual Report" series[83–95]. The U.S. epidemiologically weighted cancer mutation rates (based on ICD-O-3 and ICD-10), which integrated the data from SEER database and cBioPortal database are derived from the Supplementary files of https://www.nature.com/articles/s41467-021-26213-y[15]. All the output data in this study can be found within the Supplementary Information and Supplementary Data. The source data for all figures are provided with this paper as Source Data Files. Source data are provided with this paper.

## Code availability

The source code used to combine genomics and epidemiological data and produce all these figures and table in the manuscript is available and accessible in the Supplementary Software 1 and could also be obtained from the public code repository Zenodo [https://doi.org/10.5281/zenodo.7063609][96].

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

## Acknowledgements

The authors thank Prof. Xiang Li and Prof. Kangdong Liu for helpful comments and suggestions. This work was supported by the National Natural Science Foundation of China NSFC82073075 (Z.D.). We thank Prof. Rongshou Zheng, associate editor of "China Cancer Registry Annual Report", for the help in cancer epidemiological data recalibration.

## Author contributions

Z.D. designed the project. F.M. and K.L. collected the epidemiological and cancer sequencing data to a shared alternative cancer classification. F.M. and K.L. aggregated and processed cancer genomic data. F.M. and K.L. wrote the manuscript. K.L. and F.M. analyzed the epidemiological data and developed the computational processes for calculating the weighted mutation rates of cancer genes by combining the epidemiological data with the genomic data.

## Competing interests

The authors declare no competing interests.
