## [Peer Review File · Nature Communications]

Reviewers' comments:

Reviewer #1 (Remarks to the Author): expert in epidemiology and population statistics

This is a well-written and thorough paper that performs a genomic meta-analysis of cancer within the Chinese population and compares it with similar data from the U.S. Such epidemiological/genomic studies are likely to produce high value for offering a better understanding of cancer and for helping countries prioritize health and research priorities that offer the broadest possible value.

The comparisons between 20 specific cancer types (i.e. colon cancer in China and in U.S.) in Figure 3 is an easy, direct, comparison that quickly highlights that there are non-trivial and potentially clinically important differences between the cancers observed in the different populations.

My questions involve digging deeper to ensure that other comparisons can be fairly made between China and U.S. cancer data.

For epidemiological reweighting, the authors should better describe their coverage of the Chinese cancer epidemiological data. Surely, they do not have representative samples for every type of cancer. So how are they approximating? Or is their ICD-10 so coarse grained that "everything" is represented, because one is assuming the common cancers are good approximations of the rare cancers?

Figure 2: comparisons between U.S. and China for pan-cancer, epidemiologically weighted. The authors need to better explain what is being compared, and whether there could be artifacts in the data. In A & B of Fig 2, the authors are comparing their results (CN), which are epidemiologically re-weighted, with TCGA, which is not. Are the differences due to (a) US vs China, (b) epidemiologically weighting vs. not epidemiological weighting, or (c) differences in which specific cancers were sequenced (i.e. are there gaps in which tumors have been sequenced and are going into the respective studies). There are clearly differences and correlations, regardless, but what it means depends upon these things.

In 2C&D, the authors turn to an epidemiologically weighted estimate for the U.S., so it is a more fair comparison. However, the U.S. estimates were fine-grained epidemiologically reweighted dataset based on ICD-O-3 codes, while the China population reweighting seems to be based on much simpler, much more coarse-grained, ICD-10 codes. Could these methodological differences contribute to the differences? AI

Once the focus is on individual cancer types, these differences seem less important. (Of course, there can still be biases in which tumors are selected for sequencing, etc. - but the epidemiological reweighting is no longer at play). However, this part seems to be a simpler direct comparison of data. Additionally, some of these trends (EGFR mutations being more common in individuals of Asian heritage) are well known.

In this authors opinion, this manuscript has 3 potentially exciting areas, but none are quite finished or realized. (A) epidemiological reweighting (Figure 1 and 2) - the limitations are described above; it's not clear to the reviewer that these comparisons are meaningful because the data may have other, trivial, reasons for differences. A more thorough characterization and comparison would significantly improve this portion of the manuscript. (B) Within-cancer type comparisons between populations (Fig 3/4/5) - the idea of comparing specific cancers from different populations has less problems than (A). (However, on deep dive into cancer genomics studies - one will find that there are differences in what is classified as a form of that cancer. For example, "Pancreatic Adenocarcinoma" - some studies will include pancreatic neuroendocrine carcinomas as adenocarcinoma, while others will focus only on pancreatic ductal adenocarcinoma and have no neuroendocrine carcinomas. Thus, even these comparisons require the authors to be more thorough and to better communicate that their comparisons are between equivalent sets of cancers. (C) The signatures could be very interesting. This reflects some very interesting environmental impacts. The authors speculate that differences may reflect "continuous exposure to

mutagens or radioactive agents". That speculation could be explored, presumably, with epidemiological studies comparing where these cancers originate and if there are more exposures to radioactivity in those regions. Overall, this reviewer feels like each of (A), (B), (C) are not quite yet a fully worked out body of work, and that the combination of A+B+C is not a full body of work, either.

Reviewer #2 (Remarks to the Author): expert in population-level genomics and bioinformatics

In this study, Fayang Ma et al compare rates of gene mutations and mutational signatures across different types of cancer between two cohorts, Chinese and US patients, and identify a number similarities and some differences they attribute to various factors such as lifestyle differences and genetics. I commend the authors for a clear study with informative figures and tables. In my opinion, such a study is useful for delineating general trends, but would be wary of concluding what the observed differences may be due to without first ensuring that both cohorts and methods of analyzing these are comparable.

These are some comments that I think should be addressed (or perhaps just more extensively explained in the text) before publication:

1. I don't understand how the four sources for epidemiological data were integrated - surely the four responsible institutions have different methods? How can we be sure that data are comparable among them? Was an analysis done to ensure that's the case? For example, the cases from the four sources are summed up in Supp Table 1. I assume the data from 1983-2012 is the total of new cases in that period. Isn't it strange that there are many more cases generally in 2013 and 2014 individually than in a period of nearly 30 years immediately previous to that? Or is this the annual incidence rate and numbers need to be somehow adjusted for the period under study?
2. Can we conclude that both cohorts represent "US" and "Chinese" populations, or are these from very specific sites in both countries where certain environmental or genetic factors may play a role in cancer risk?
3. What is the rationale for ensuring that a specific set of genomic samples (the ones taken from cBioPortal) is generalizable to the data from various sites of China considered in the epidemiology analysis? For example, for the mutation signatures analysis that comes later, is it important to consider what proportions of patients in both cohorts have undergone treatment (this would affect mutational signature calling)?
4. In Section 2, "Differences in the mutation rates of the top 50 cancer genes between Chinese and U.S. patient populations", is the positive correlation observed due in great part to TP53? How does the R2 change if TP53 is removed from the dataset?
5. I would be worried about reaching conclusions in the comparison of the lower end of the top 10 mutated genes in US vs China (the 10th gene has a mutation prevalence 0.4 in the Chinese dataset and 0.3 in the US). How can we be sure that mutation calling, gene filtering, qualities etc were equally applied between both datasets?
6. In Section 4, "Mutation rates of top 50 genes in each of the 20 cancer types were compared between China and U.S.", can it be specified what a conservation rate is? When it says that LUAD has a 68% conservation rate, what does it mean - that 68% of top 50 genes are shared between both cohorts? Is this meaningful - would it change if another arbitrary number of genes was chosen, say 20 or 100?
7. Can the PCLO discrepancies be explained by anything else than demographics? For example, that in one pipeline it had lower coverage, or is it in a difficult region to call variants? I think these hypotheses need to be discarded before assuming it is due to something population-specific.
8. I would be worried about concluding SBS60 is a real signature observed in Chinese samples, as is classified as a sequencing artifact in COSMIC. This goes again to the point about variant calling

and filtering differences between cohorts.

9. In Methods, "Mutation Proportion Estimates", it says "Within the present study, the number of genes (m) is 1,043 and the number of cancer types (n) is 20." Where does this number come from? In Results, section "The prevalence rates of mutated cancer genes observed across Chinese cancer patients", it says that 1,064 genes were considered and that 1,036 genes were mutated.

10. Please substitute the word Caucasian for "European-descent" or something similar - Caucasian is considered and outdated and racist term.

Reviewer #1 (Remarks to the Author): expert in epidemiology and population statistics

This is a well-written and thorough paper that performs a genomic meta-analysis of cancer within the Chinese population and compares it with similar data from the U.S. Such epidemiological/genomic studies are likely to produce high value for offering a better understanding of cancer and for helping countries prioritize health and research priorities that offer the broadest possible value.

Answer: Thank you for your highly positive comments.

According to your constructive suggestions and comments, we did a thorough re-analysis, well addressed all the concerns, and substantially increased the reliability and accuracy of all the methodologies and results. Which were presented in five major aspects:

1). The national cancer incidence data used for epidemiologically reweighting of the mutation rates of cancer genes were derived from the series of "China Cancer Registry Annual Report" composed and published by Chinese National Cancer Center (NCC) from 2004 to 2019 (Data range, 2004~2016), which were proved to be with high quality and accuracy;

2). The U.S. mutation rates of cancer genes were epidemiologically reweighted based on ICD-10 which were recalculated from reclassifying the ICD-O-3 classification system, to ensure the CN vs U.S. comparison on the same standard. Further, the two sets of mutation rates (U.S.) epidemiologically weighted based on ICD-O-3 and ICD-10 were compared and showed very high consistency and no significant differences;

3). To avoid bias, incompatibility, and inconsistency produced in the process of integrating different sequencing data together for comparison, we performed a quality control test by using four most frequently used variant callers (MuTect2, VarScan2, SomaticSniper, and Muse) in the analysis of one example of sequencing data, and the results showed high consistency and showed no significant differences among different mutation callers. Which implicated that the reliability of methodology and accuracy of results were reasonably ensured in the comparison;

4). To ensure the compatibility and comparability of integrating multiple sequencing data and comparison between CN and U.S. cohorts, we use a panel of 382 cancer gene census from OncoKB database and same mutation classifications to filter all mutational profiles from both cohorts. Additionally, 45 cancer subtypes across the 23 tumor sites in the comparison were commonly shared by CN and U.S cohorts. The case number of which were 91.6% and 86.2% of all the CN and U.S. cohorts, respectively.

5). In the comparison of mutational signatures between CN and U.S. cohorts, only tumor samples with treatment-naïve status were included for the corresponding analysis to ensure the mutation patterns were not affected by a diversity of treatment options.

1. *For epidemiological reweighting, the authors should better describe their coverage of the Chinese cancer epidemiological data. Surely, they do not have representative samples for every type of cancer. So how are they approximating? Or is their ICD-10 so coarse grained that "everything" is represented, because one is assuming the common cancers are good approximations of the rare cancers?*

Answer: We agree that it is essentially important to have a sufficient coverage of the Chinese cancer epidemiological data with good quality. The previous used epidemiological data of cancer patients between 1983~2020 were derived from four sources, and we found it is with less completeness and less comparability, and may not be well integrated together. Now, we have

replaced these data with the annual cancer incidence from 2004 to 2016, which were derived from the consecutive series of China Cancer Registry Annual Report composed and published by Chinese National Cancer Center (NCC). These Cancer registries collected data on all cancers' incidence, mortality, and survival. The demographic information and diagnostic information of the registered cancer cases were also recorded. In the meantime, the data of population coverage as the census data would be collected from departments of statistics and public security. Cancer incidence would be recorded and reclassified based on the ICD-10 classification system. After receiving the cancer registration data, NCC would use the IARC/IACR-check software to evaluate the completeness, validity, and internal consistency of the data. The cancer datasets would be examined and revised based on the evaluation results. Finally, qualified cancer datasets would be pooled and published as China Cancer Registry Annual Report. In the 13 years' range from 2004~2016, 5,878,712 cancer patients has been registered. For instance, a total of 1,110,867 new cancer patients registered in 487 qualified cancer registries were included in the "China Cancer Registry Annual Report, 2019" (2016 data), which covered a total of 381,565,422 population (193,632,323 males, 187,933,099 females), accounting for 27.60% of the national population in 2016.

According to the "Guidelines of Chinese Cancer Registration", "Technical Protocols of Cancer Registration and Follow Up", and the standards of International Agency for Research on Cancer/International Association of Cancer Registries (IARC/IACR) on "Cancer Incidence in Five Continents, Vol. XI", a national criterion to evaluate the quality of Chinese cancer registration data has been established following the rules of comparability, completeness, validity, and timeliness. The detailed method used for data collection in cancer registries, including Methods and Index, Methods of Data Collection, Channels of Data Collection, Certification of Cancer Cases, and Follow-up Practice, could be referred to the series of "China Cancer Registry Annual Report" composed and published by Chinese National Cancer Center (NCC) from 2004 to 2019 (Data range, 2004~2016).

The new epidemiological data approximately grained everything. We believe that our study could largely represent the actual mutation rates of cancer genes in Chinese cancer population after being reweighted by epidemiological data based on the ICD-10 classification system. We should thank Prof. Rongshou Zheng, associate editor of "China Cancer Registry Annual Report", for the help in cancer data recalibration.

2. *Figure 2: comparisons between U.S. and China for pan-cancer, epidemiologically weighted. The authors need to better explain what is being compared, and whether there could be artifacts in the data. In A & B of Fig 2, the authors are comparing their results (CN), which are epidemiologically re-weighted, with TCGA, which is not. Are the differences due to (a) US vs China, (b) epidemiologically weighting vs. not epidemiological weighting, or (c) differences in which specific cancers were sequenced (i.e. are there gaps in which tumors have been sequenced and are going into the respective studies). There are clearly differences and correlations, regardless, but what it means depends upon these things.*

Answer: Thank you for your suggestion. We provided the list of all the cancer types in both cohorts (Supplemental file_Sequencing data_Cancer types_ICD10), and 91.6% of CN cohorts and 86.2% of U.S. cohorts fell into the same 45 cancer subtypes in 23 tumor sites (ICD-10 classification) (S. Table 1), which were utilized to compare the epidemiologically weighted mutation rates of cancer genes between U.S. and China, and also the possible limitations. According to your comments, TCGA is not epidemiologically re-weighted. Therefore, we delete all

the results produced by comparing with TCGA.

3. *In 2C&D, the authors turn to an epidemiologically weighted estimate for the U.S., so it is a more fair comparison. However, the U.S. estimates were fine-grained epidemiologically reweighted dataset based on ICD-O-3 codes, while the China population reweighting seems to be based on much simpler, much more coarse-grained, ICD-10 codes. Could these methodological differences contribute to the differences?*

Answer: We agree with you. In order to eradicate the potential differences attributed from the methodological differences between ICD-O-3 (U.S.) vs ICD-10 (CN). We turn to reclassify the U.S. data (ICD-O-3) using ICD-10 classification system, which guarantees the accuracy and reliability of the comparison via the same classification system. Further, the two sets of mutation rates (U.S.) epidemiologically weighted based on ICD-O-3 and ICD-10 were compared and showed very high consistency and no significant differences (S.Figure 4).

4. *Once the focus is on individual cancer types, these differences seem less important. (Of course, there can still be biases in which tumors are selected for sequencing, etc. - but the epidemiological reweighting is no longer at play). However, this part seems to be a simpler direct comparison of data. Additionally, some of these trends (EGFR mutations being more common in individuals of Asian heritage) are well known.*

Answer: Thank you for your positive comments. The trend that *EGFR* mutations are more common in individuals of Asian heritage is again being presented and confirmed in our results. In the meantime, this consistency as a positive control implicates the accuracy and reliability of other trends presented in the results.

5. *In this authors opinion, this manuscript has 3 potentially exciting areas, but none are quite finished or realized.*

(A) *epidemiological reweighting (Figure 1 and 2) - the limitations are described above; it's not clear to the reviewer that these comparisons are meaningful because the data may have other, trivial, reasons for differences. A more thorough characterization and comparison would significantly improve this portion of the manuscript.*

(B) *Within-cancer type comparisons between populations (Fig 3/4/5) - the idea of comparing specific cancers from different populations has less problems than (A). (However, on deep dive into cancer genomics studies - one will find that there are differences in what is classified as a form of that cancer. For example, "Pancreatic Adenocarcinoma" - some studies will include pancreatic neuroendocrine carcinomas as adenocarcinoma, while others will focus only on pancreatic ductal adenocarcinoma and have no neuroendocrine carcinomas. Thus, even these comparisons require the authors to be more thorough and to better communicate that their comparisons are between equivalent sets of cancers.*

(C) *The signatures could be very interesting. This reflects some very interesting environmental impacts. The authors speculate that differences may reflect "continuous exposure to mutagens or radioactive agents". That speculation could be explored, presumably, with epidemiological studies comparing where these cancers originate and if there are more exposures to radioactivity in those regions. Overall, this reviewer feels like each of (A), (B), (C) are not quite yet a fully worked out body of work, and that the combination of A+B+C is not a full body of work, either.*

A_Answer: We thank you for your important question on the epidemiological reweighting. For the CN cancer epidemiological data, a more thorough characterization has been provided based on high quality data with completeness and comparability derived from a consecutive series of China National Cancer Registry Report (2004~2016). For the comparison between U.S. and CN, the mutation rates of U.S. cohort were recalculated and epidemiologically reweighted based on the ICD-10 classification system. Consistently, the mutation rates produced from ICD-O-3 and

ICD-10 were largely similar and showed no significant differences. Additionally, 45 cancer subtypes across the 23 tumor sites being compared were commonly shared by CN and U.S cohorts. The case number of which were 91.6% and 86.2% of all the CN and U.S. cohorts, respectively.

B_Answer: Thank you for your positive comments. For thorough comparison between equivalent sets of cancers, standardization was performed by two pathologists via manually examining the histological information of all cancer types in both U.S. and CN cohorts. The carcinomas under "Pancreatic Adenocarcinoma" include pancreatic ductal adenocarcinoma, but do not include any neuroendocrine carcinomas. The equivalent sets of cancers for comparison were provided in S.Table 1.

C_Answer: Thank you for your valuable suggestions. After a thorough examination of the clinical information of the patients, only samples with treatment-naïve status were retained for mutation signature analysis. The mutational signatures for papillary thyroid carcinoma (CN/U.S.), skin cutaneous melanoma(CN/U.S.), and lung adenocarcinoma(female nonsmokers, CN/U.S.) were reanalyzed as SBS25, SBS7, and SBS40, respectively. The only minor differences were the mutational signatures between CN-SKCM (SBS7b) and U.S.-SKCM (SBS7a), However, the proposed etiology of the two signatures both fell in the category of ultraviolet radiation exposure, which has been associated with DNA replication timing (30). Additionally, SBS7a/b were mutually exclusive with the normal aging related mutational signature SBS1/5 (31, 32).

Reviewer #2 (Remarks to the Author): expert in population-level genomics and bioinformatics

In this study, Fayang Ma et al compare rates of gene mutations and mutational signatures across different types of cancer between two cohorts, Chinese and US patients, and identify a number similarities and some differences they attribute to various factors such as lifestyle differences and genetics. I commend the authors for a clear study with informative figures and tables. In my opinion, such a study is useful for delineating general trends, but would be wary of concluding what the observed differences may be due to without first ensuring that both cohorts and methods of analyzing these are comparable.

These are some comments that I think should be addressed (or perhaps just more extensively explained in the text) before publication:

Answer: Thank you for your positive comments. These problems have been well addressed to ensure that both cohorts and methods of analyzing these are comparable and with higher reliability.

Additionally, according to your constructive suggestions and comments, we did a thorough re-analysis, well addressed all the concerns, and substantially increased the reliability and accuracy of all the methodologies and results. Which were presented in five major aspects:

1). The national cancer incidence data used for epidemiologically reweighting of the mutation rates of cancer genes were derived from the series of "China Cancer Registry Annual Report" composed and published by Chinese National Cancer Center (NCC) from 2004 to 2019 (Data range, 2004~2016), which were proved to be with high quality and accuracy;

2). The U.S. mutation rates of cancer genes were epidemiologically reweighted based on ICD-10 which were recalculated from reclassifying the ICD-O-3 classification system, to ensure the CN vs U.S. comparison on the same standard. Further, the two sets of mutation rates (U.S.) epidemiologically weighted based on ICD-O-3 and ICD-10 were compared and showed very high consistency and no significant differences;

3). To avoid bias, incompatibility, and inconsistency produced in the process of integrating different sequencing data together for comparison, we performed a quality control test by using four most frequently used variant callers (MuTect2, Varscan2, SomaticSniper, and Muse) in the analysis of one example of sequencing data, and the results showed high consistency and showed no significant differences among different mutation callers. Which implicated that the reliability of methodology and accuracy of results were reasonably ensured in the comparison;

4). To ensure the compatibility and comparability of integrating multiple sequencing data and comparison between CN and U.S. cohorts, we use a panel of 382 cancer gene census from OncoKB database and same mutation classifications to filter all mutational profiles from both cohorts. Additionally, 45 cancer subtypes across the 23 tumor sites in the comparison were commonly shared by CN and U.S cohorts. The case number of which were 91.6% and 86.2% of all the CN and U.S. cohorts, respectively.

5). In the comparison of mutational signatures between CN and U.S. cohorts, only tumor samples with treatment-naïve status were included for the corresponding analysis to ensure the mutation patterns were not affected by a diversity of treatment options.

1. *I don't understand how the four sources for epidemiological data were integrated - surely the four responsible institutions have different methods? How can we be sure that data are comparable among them? Was an analysis done to ensure that's the case? For example, the*

cases from the four sources are summed up in Supp Table 1. I assume the data from 1983-2012 is the total of new cases in that period. Isn't it strange that there are many more cases generally in 2013 and 2014 individually than in a period of nearly 30 years immediately previous to that? Or is this the annual incidence rate and numbers need to be somehow adjusted for the period under study?

Answer: Thank you for your valuable comments. We realized that the previous used epidemiological data from the four distinct sources were with less completeness and less comparability, and may not be well integrated together.

Now, we have replaced these epidemiological data with the qualified Chinese annual cancer incidence from 2004 to 2016, which were derived from the consecutive series of China Cancer Registry Annual Report composed and published by Chinese National Cancer Center (NCC). These Cancer registries collected data on all cancers' incidence, mortality, and survival. The demographic information and diagnostic information of the registered cancer cases were also recorded. In the meantime, the data of population coverage as the census data would be collected from departments of statistics and public security. Cancer incidence would be recorded and reclassified based on the ICD-10 classification system. After receiving the cancer registration data, NCC would use the IARC/IACR-check software to evaluate the completeness, validity, and internal consistency of the data. The cancer datasets would be examined and revised based on the evaluation results. Finally, qualified cancer datasets would be pooled and published as China Cancer Registry Annual Report. In the 13 years' range from 2004~2016, a total of 5,878,712 cancer patients has been registered. For instance, a total of 1,110,867 new cancer patients registered in 487 qualified cancer registries were included in the "China Cancer Registry Annual Report, 2019" (2016 data), which covered a total of 381,565,422 population (193,632,323 males, 187,933,099 females), accounting for 27.60% of the national population in 2016.

According to the "Guidelines of Chinese Cancer Registration", "Technical Protocols of Cancer Registration and Follow Up", and the standards of International Agency for Research on Cancer/International Association of Cancer Registries (IARC/IACR) on "Cancer Incidence in Five Continents, Vol. XI", a national criterion to evaluate the quality of Chinese cancer registration data has been established following the rules of comparability, completeness, validity, and timeliness. The detailed method used for data collection in cancer registries, including Methods and Index, Methods of Data Collection, Channels of Data Collection, Certification of Cancer Cases, and Follow-up Practice, could be referred to the series of "China Cancer Registry Annual Report" composed and published by Chinese National Cancer Center (NCC) from 2004 to 2019 (Data range, 2004~2016). We should thank Prof. Rongshou Zheng, associate editor of "China Cancer Registry Annual Report", for the help in cancer data recalibration.

Our analysis showed that the cancer incidence numbers in each of the 13 years' epidemiological data (2004~2016), and the corresponding annual population covered have a highly positive correlated rate ($R^2=1.00$) (S.Figure 2B~2C). So we believed that data of consistency, comparability, and reliability of integration are ensured via the new data.

2. *Can we conclude that both cohorts represent "US" and "Chinese" populations, or are these from very specific sites in both countries where certain environmental or genetic factors may play a role in cancer risk?*

Answer: Yes, we conclude that both cohorts could well represent U.S. and CN populations. The epidemiological and demographic information of both cohort patients, showed that these samples were collected from multiple areas which widely distributed across the CN and U.S., respectively

(S. Figure 2A) (Supplemental file_Epidemiological Data_3,5). So we conclude that both cohorts generally represent CN and U.S. population, with respect to both sequencing data and epidemiological data for weighting mutation rates.

3. *What is the rationale for ensuring that a specific set of genomic samples (the ones taken from cBioPortal) is generalizable to the data from various sites of China considered in the epidemiology analysis? For example, for the mutation signatures analysis that comes later, is it important to consider what proportions of patients in both cohorts have undergone treatment (this would affect mutational signature calling)?*

Answer: Thank you for your question. According to the national cancer epidemiological data, the corresponding genomic data were derived from multiple sources, including cBioPortal (China Pan-cancer project_Origimed2020), ICGC database, and published genomic studies (details in Method). These tumor samples were collected from areas and places, which widely distributed across China (S. Figure 2A) (Supplemental file_Epidemiological Data_3,5). We concluded that the genomic samples (CN_11,948, U.S._18564) and epidemiological data for reweighting mutation rates could basically be generalizable to the various sites of CN and U.S. in the study.

We agree with your comment on the treatment status (especially the targeted therapy), which would significantly affect the mutation signature calling. We performed a thorough examination of the corresponding clinical data of all these samples selected for mutation signature analysis. Only samples with the status of treatment naïve are retained for mutational signature calling. The detailed clinical treatment status for CN/U.S._LUAD, SKCM, and PTC were provided in the Supplemental file_Treatment status data.

4. *In Section 2, "Differences in the mutation rates of the top 50 cancer genes between Chinese and U.S. patient populations", is the positive correlation observed due in great part to TP53? How does the R² change if TP53 is removed from the dataset?*

Answer: Thank you for your suggestion. We noticed that the R^2 would be decreased from 0.91 to 0.87 when a good contribution of *TP53* is removed (Figure. 2C). Which indicated the correlation of mutation rates of cancer genes between Chinese and U.S. patient populations is still on the positive side.

5. *I would be worried about reaching conclusions in the comparison of the lower end of the top 10 mutated genes in US vs China (the 10th gene has a mutation prevalence 0.4 in the Chinese dataset and 0.3 in the U.S). How can we be sure that mutation calling, gene filtering, qualities etc were equally applied between both datasets?*

Answer: Thank you for your comments. The lower end of the top10 mutated genes were removed. As our goal was to tabulate the proportion of cancer patients who have a cancer gene mutated, we counted the patient sample as mutated whether it had a single mutation or multiple mutations within that same gene. In the meantime, we performed a quality control test by using four different variant callers (MuTect2, VarScan2, SomaticSniper, and Muse) to process the same example (raw data), and the results showed high consistency and showed no significant differences among different mutation calling methods (S. Figure 3). Which implicated that the reliability of methodology and accuracy of results were reasonably ensured in the comparison.

6. *In Section 4, "Mutation rates of top 50 genes in each of the 20 cancer types were compared between China and U.S.", can it be specified what a conservation rate is? When it says that LUAD has a 68% conservation rate, what does it mean - that 68% of top 50 genes are shared between both cohorts? Is this meaningful - would it change if another arbitrary number of genes was*

chosen, say 20 or 100?

Answer: We agree with you on that. The conservation rate has been replaced by common rates in the context of top50 genes.

7. *Can the PCLO discrepancies be explained by anything else than demographics? For example, that in one pipeline it had lower coverage, or is it in a difficult region to call variants? I think these hypotheses need to be discarded before assuming it is due to something population-specific.*

Answer: Thank you for your suggestion. The portion of PCLO was no longer existed after a thorough major revision.

8. *I would be worried about concluding SBS60 is a real signature observed in Chinese samples, as is classified as a sequencing artifact in COSMIC. This goes again to the point about variant calling and filtering differences between cohorts.*

Answer: Thank you for your suggestion. The portion of mutational signature SBS60, classified as a possible sequencing artifact in COSMIC, has been deleted to avoid causing unnecessary confusions.

REVIEWER COMMENTS

Reviewer #1 (Remarks to the Author):

The authors revised manuscript includes many changes that were made in response to the specific requests from the original review. The changes satisfy the general spirit of the questions and are acceptable. They improve the manuscript and its ability to address its key messages.

The last comment of my original review stated that I was not sure if the three components of the study combined to make a complete advance. I do believe the changes improve the overall quality of the manuscript, and allow the authors to do what they set out to do. The editors now get to decide whether the manuscript fits the vision for the journal.

Reviewer #2 (Remarks to the Author):

Thanks to the authors for their thoughtful responses. I can appreciate the amount of work that went into this revision. I do think the new data on Chinese cancer patients is much clearer and easier to understand, thank you for addressing this issue. I am sorry if I still have some minor questions regarding the new methodology:

1. How was the "population covered/100,000" calculated (Supp Figs 3B and 3C)? In Epidemiological data_4, for 2016 (year of publication 2019), it says that there were 381,565,422 people covered from a total of 1,382,483,413 individuals (27%). Wouldn't this mean that 27/100, or 27,000/100,000 population were covered? Where does the 3,816 come from?

2. Also, on the same lines, how was the total number of 2,069,289,220 individuals derived? Is the figure 11 cancer patients per 100,000 too low? The US has an incidence of about 40 times that number. I apologize if I have not understood these numbers and calculations, but could they be further clarified in the text please?

3. My question 2 in the first round of revisions, regarding whether the cohorts can be taken as representative of the US/Chinese populations, may require further thought than just showing that patients are recruited from many different sites. For example, are the great majority of patients in the US registry from European descent, as these patients can generally more easily access health centres? Can that ancestry be projected onto the rest of the country? Or do the authors not care about genetic ancestry influences on genomic profiles of cancer, and if so, what would be their justification?

4. Linked to the above, for their results (for example the differences in rates of mutation of EGFR and KRAS between Chinese and US lung adenocarcinoma patients) have the authors made sure that these differences do not come from a specific mutational profile in a single site and are indeed spread across all collection sites? If these are indeed spread out - what does it mean (e.g. does ethnicity have anything to do with these differences? Does an environment shared across the country?)

5. For the Methods, could this be rephrased please? "However, as our goal was to tabulate the proportion of cancer patients who have a gene mutated (or not), we flagged a given gene within a specific patient sample as mutated if the individual mutation observed within that gene exceeded one. " At the moment I do not understand what the "one" refers to - one read? One individual mutation?

6. Given that the authors did find differences in the different variant calling methods, have they considered whether any of their results could be false positives due to these differences?

7. For the mutational signatures analysis, can the observed differences be attributed to the difference in sequencing coverage (e.g., MSK-IMPACT has sequencing of only a panel of genes whereas other sources have whole exomes)? It is known that the accuracy of mutational signature

calling increases with genomic space considered, so, could these differences not play a role in the results?

Minor points

1. In line 104, it says "The top mutated cancer gene is TP53 both in China (51.4%) and U.S. (33.7%), with a 17.7% prominent difference between the two population groups. " I would remove the 17.7% as it is not clear what it is referring to (It seems like a subtraction but as it is a percentage it can be misunderstood).

2. In line 128, "Other cancer types have distinct top mutated genes; KRAS for pancreatic adenocarcinoma (PAAD) in CN & U.S...." but KRAS is the same top mutated gene in both US and Chinese PAAD cohorts? As it is written I do not understand this sentence.

Reviewer #1 (Remarks to the Author):

The authors revised manuscript includes many changes that were made in response to the specific requests from the original review. The changes satisfy the general spirit of the questions and are acceptable. They improve the manuscript and its ability to address its key messages.

The last comment of my original review stated that I was not sure if the three components of the study combined to make a complete advance. I do believe the changes improve the overall quality of the manuscript, and allow the authors to do what they set out to do. The editors now get to decide whether the manuscript fits the vision for the journal.

Answer: Thank you. Your previous suggestions and comments greatly contribute to the improvement of the overall quality of the manuscript, which ensure the reliability of the data and the accuracy of the results. We truly appreciate it.

Reviewer #2 (Remarks to the Author):

Thanks to the authors for their thoughtful responses. I can appreciate the amount of work that went into this revision. I do think the new data on Chinese cancer patients is much clearer and easier to understand, thank you for addressing this issue. I am sorry if I still have some minor questions regarding the new methodology:

1. How was the "population covered/100,000" calculated (Supp Fig. 2b and 2c)? In Epidemiological data_4, for 2016 (year of publication 2019), it says that there were 381,565,422 people covered from a total of 1,382,483,413 individuals (27%). Wouldn't this mean that 27/100, or 27,000/100,000 population were covered? Where does the 3,816 come from?

Answer: Thank you for your question. Please refer to Supp Fig. 2b and 2c. The "Population Covered/100,000" was the exact value cited from the Statistics of Cancer Registry data (2004~2016). The "3,816" under the entity "Population Covered/100,000" for 2016 is the exact value 381,565,422 (Population Covered), calculated via $381,565,422/100,000$. ' / ' in the "Population Covered/100,000" means 'divided by', not 'per' or 'in', only for the purpose of a concise presentation in the graph. "The "Population Covered" for 2016 means that 1,110,867 patients (No of Registered Cancer Patients) were represented within 381,565,422 people (Population Covered); 381,565,422 (Population Covered) is around 27.6% of the Total Population_CN (1,382,483,413). We are sorry for the confusion caused by misusing of ' / ', which has been removed to avoid the ambiguous meaning.

2. Also, on the same lines, how was the total number of 2,069,289,220 individuals derived? Is the figure 11 cancer patients per 100,000 too low? The US has an incidence of about 40 times that number. I apologize if I have not understood these numbers and calculations, but could they be further clarified in the text please?

Answer: Thank you for your question. 2,069,289,220 is the sum of the “ Population Covered ” between 2004~2016 (in replicated counting), that number has been removed to avoid unnecessary misunderstanding. ‘ / ’ in the “Population Covered/100,000” means divided by, not ‘per’ or ‘in’. As noted above. We are sorry for the confusion caused by misusing of ‘/’, which has been removed to avoid the ambiguous meaning and potential misunderstanding.

3. My question 2 in the first round of revisions, regarding whether the cohorts can be taken as representative of the US/Chinese populations, may require further thought than just showing that patients are recruited from many different sites. For example, are the great majority of patients in the US registry from European descent, as these patients can generally more easily access health centres? Can that ancestry be projected onto the rest of the country? Or do the authors not care about genetic ancestry influences on genomic profiles of cancer, and if so, what would be their justification?

Answer: Thank you for your suggestions. For CN cohort, Han Chinese comprised 91.59%, 91.51%, and 91.11% of the total population in 2000, 2010, and 2020, respectively (Data from National Bureau of Statistics, <http://www.stats.gov.cn/tjsj/tjgb/rkpcgb/>). Additionally, the proportions of urban population in the total population were documented as 36.22%, 49.68%, and 63.89% in 2000, 2010, and 2020, respectively (Data from National Bureau of Statistics) (S. Fig. 2d). Over the past decades, the increased speed of urbanization and population flow at the national level greatly increases genomic hybridization and homogenization in China (S. Fig. 2e). Therefore, we conclude that the cohort of 11,948 cancer patients recruited from multiple areas is generally representative of the Chinese cancer population with respect to the ancestry background.

With respect to the U.S., we are indeed aware that specific cultural and socio-economic factors play a role in patient representation. According to the race and ethnicity statistics represented within the SEER registry (<https://seer.cancer.gov/statistics-network>). Delay-adjusted incidence rates for European Descent and African Descent appear comparable over a 19 year period. An increase in the number of American-Indian/Alaskan Native is apparent, while a decrease can be seen for the Hispanic population group. A flat-trend is observed for members of the Asian/Pacific Islander group. Importantly, the statistical analyses presented in the SEER database utilize a technique known as race bridging which allows data collected using one set of race categories consistent with data collected using a different set of race categories. The bridged-race estimates used within the study were produced under a collaborative arrangement with the U.S. Census Bureau. Corresponding details could be found at https://www.cdc.gov/nchs/nvss/bridged_race.htm. The ancestry influences on genomic profiles have also been considered and provided for the included U.S. cohorts. From the 139 studies included in the U.S. cohorts, 11,526 cancer patients with a variety of known ancestry background were presented in the pie chart (S. Fig. 2f). Your suggestions for improving the reliability and rationale of this work are appreciated.

4. Linked to the above, for their results (for example the differences in rates of mutation of EGFR and KRAS between Chinese and US lung adenocarcinoma patients) have the authors made sure that these differences do not come from a specific mutational profile in a single site and are indeed spread across all collection sites? If these are indeed spread out - what does it mean (e.g. does ethnicity have anything to do with these differences? Does an environment shared across the country?)

Answer: Thank you for your comments. In the current analysis, the *EGFR*^{L858R} mutation was identified in 45.2% of 478 Chinese LUAD_female non-smokers. The higher *EGFR* mutation rate is consistent with many other studies

conducted in Asian NSCLC patients, in which 40~55% patients have TKI-sensitive mutations, with *EGFR*^{L858R} and *EGFR*^{Ex19Del} comprising most cases. In contrast, 16.4% of U.S. female TCGA-LUAD patients have *EGFR* mutations, and only 6.2% of these patients harbor the *EGFR*^{L858R} mutation.

The potential etiological or environmental factors underlying these differences of mutation rates between two countries were investigated through mutational signature analysis. The results show that LUAD-CN female non-smokers and LUAD-U.S. female non-smokers share the same etiology mutation signature (SBS40) (Fig. 6a, 6b). In several Asian populations, including China, an increased risk of female lung cancer is likely associated with indoor pollution produced by heating and cooking oil fumes. The evidence above suggests that the LUAD-female non-smokers from both U.S. and CN are potentially exposed to the same environmental carcinogens- yet exposure may occur at increased frequency in China.

Similar phenomena also exist for papillary thyroid carcinoma (PTC) and skin cutaneous melanoma (SKCM) in U.S. and CN cohorts. Though there are differences in the mutation rates (Fig. 4c), the underlying etiological factors are the same, SBS25 for both PTC cohorts, and SBS7 for both SKCM cohorts, respectively. Based on the above evidences, we could conclude that these mutation rates differences and the underlying environmental risk factors could spread out nationally and be comparable between CN and U.S.

5. For the Methods, could this be rephrased please? "However, as our goal was to tabulate the proportion of cancer patients who have a gene mutated (or not), we flagged a given gene within a specific patient sample as mutated if the individual mutation observed within that gene exceeded one. " At the moment I do not understand what the "one" refers to - one read? One individual mutation?

Answer: Thank you for your suggestion. "one" refers to one individual mutation. That sentence has been revised to be clearer as "Our main focus was to calculate the proportion of cancer patients harboring a specific mutated gene (from a panel of 382 cancer genes) in one cohort, the cancer patient would be counted if one or more harbored mutations fall into our established category". Thank you for your suggestion to make that sentence easier to understand.

6. Given that the authors did find differences in the different variant calling methods, have they considered whether any of their results could be false positives due to these differences?

Answer: Thank you for your question. Each individual variant caller typically provides a metric of statistical reliability (p -value or q -value) for each called variant using the Benjamini-Hochberg procedure. Variants that fail to meet the statistical threshold are withheld from further analysis.

Additionally, in the comparison between two individual cohorts, only one or two variant callers would be involved. For instance, in S. Fig. 3c, the percentage of shared mutated genes is 81.1%. If two variant callers were used in a comparison, that percentage would be greatly increased to 82.3%~94.3%. Additionally, the patient number corresponding to each of the shared mutated genes produced from the four variant callers were compared and showed no significant differences (S. Fig 3f). Most importantly, the patient number corresponding to each of the commonly shared mutated genes produced from the four callers were well overlapping with each other (S. Fig 3g). With respect to S. Fig 3g, we developed a metric to assess the maximum error exist within this particular dataset. 7.7% as an maximum error rate was resulted from calculating the sum of the absolute value of pairwise differences in observed variant call for each individual gene across the four variant callers.

Therefore, we believe that the results produced using the current methodology could well represent the real situation of mutation rates in the population level, even accompanied with small variances between different variant calling methods.

7. For the mutational signatures analysis, can the observed differences be attributed to the difference in sequencing coverage (e.g., MSK-IMPACT has sequencing of only a panel of genes whereas other sources have whole exomes)? It is known that the accuracy of mutational signature calling increases with genomic space considered, so, could these differences not play a role in the results?

Answer: Thank you for this question. We agree that the accuracy of mutational signature calling increases with genomic space considered.

In order to reduce the potential bias produced from varying sequencing coverage in the mutational signature analysis, we have applied a protein-coding mutation spectra derived from the top 50 mutated genes in each cohort. Additionally, to verify that the confined mutational spectra meets the threshold for signature identification, we queried the corresponding data derived from a cohort of U.S. lung adenocarcinoma patients with smoking history. The input of which precisely produced a mutational signature (SBS4) for smoking. Additionally, the etiological risk factor underlying the skin cutaneous melanoma produced from current mutational signature analysis accurately pointed to SBS7 in Fig. 6c~6d, which are found in skin cancer from sun exposed areas and are likely to be due to exposure to ultraviolet light.

Therefore, we believe that the equalized mutational spectra used in the current pipeline could accurately identify the mutation signatures and truly associated with the underlying etiological factors.

Minor points

1. In line 104, it says "The top mutated cancer gene is TP53 both in China (51.4%) and U.S. (33.7%), with a 17.7% prominent difference between the two population groups." I would remove the 17.7% as it is not clear what it is referring to (It seems like a subtraction but as it is a percentage it can be misunderstood).

Answer: Thank you for your considerable suggestion. The 17.7% has been removed to avoid misunderstanding. "The top mutated cancer gene is TP53 both in China (51.4%) and U.S. (33.7%)."

2. In line 128, "Other cancer types have distinct top mutated genes; KRAS for pancreatic adenocarcinoma (PAAD) in CN & U.S...." but KRAS is the same top mutated gene in both US and Chinese PAAD cohorts? As it is written I do not understand this sentence.

Answer: Thank you for your concern about the confusions. "Other cancer types have distinct top mutated genes;..." has been revised as "In most cancer types included in this study, U.S. and CN patients share the same top gene;....", which is consistent with the following text. Your careful reading through the whole manuscript greatly improved the accuracy and quality, it is appreciated.

REVIEWERS' COMMENTS

Reviewer #2 (Remarks to the Author):

Thanks to the authors for patiently answering and addressing my questions. I am happy with their answers now.